# The ascending arousal system shapes neural dynamics to mediate awareness of cognitive states

Brandon R. Munn [1,2,3✉], Eli J. Müller[1,2,3], Gabriel Wainstein [1,2] & James M. Shine [1,2✉]

Models of cognitive function typically focus on the cerebral cortex and hence overlook functional links to subcortical structures. This view does not consider the role of the highly-conserved ascending arousal system's role and the computational capacities it provides the brain. We test the hypothesis that the ascending arousal system modulates cortical neural gain to alter the low-dimensional energy landscape of cortical dynamics. Here we use spontaneous functional magnetic resonance imaging data to study phasic bursts in both locus coeruleus and basal forebrain, demonstrating precise time-locked relationships between brainstem activity, low-dimensional energy landscapes, network topology, and spatio-temporal travelling waves. We extend our analysis to a cohort of experienced meditators and demonstrate locus coeruleus-mediated network dynamics were associated with internal shifts in conscious awareness. Together, these results present a view of brain organization that highlights the ascending arousal system's role in shaping both the dynamics of the cerebral cortex and conscious awareness.

[1] Complex Systems Research Group, The University of Sydney, Sydney, NSW, Australia. [2] Brain and Mind Centre, The University of Sydney, Sydney, NSW, Australia. [3] These authors contributed equally: Brandon R. Munn, Eli J. Müller. ✉email: brandon.munn@sydney.edu.au; mac.shine@sydney.edu.au

It is often difficult to see the forest for the trees, but to fully understand a concept typically involves an accurate depiction of both. That is, we need to comprehend not only the detailed workings of a specific system, but also how that system functions within a broader context of interacting parts. Modern theories of whole-brain function exemplify this challenge. For instance, activity in the brain has been shown to incorporate signatures of both local computational specificity (e.g. specialized regions within the cerebral cortex) as well as system-wide integration (e.g. interactions between the cortex and the rest of the brain)[1,2]. Anatomical evidence suggests that the balance between integration and segregation is mediated in part by the relatively fixed white matter connections between cerebral cortical regions[1]—local connectivity motifs support segregated activity, whereas the axonal, re-entrant connections between regions act to integrate the distributed signals via a highly interconnected structural backbone[3]. However, how the human brain is also capable of remarkable contextual flexibility given this relatively fixed connectivity remains poorly understood.

During cognitive tasks, neural activity rapidly reconfigures the functional large-scale network architecture of the brain to facilitate coordination between otherwise segregated cortical regions. Precisely how this flexibility is implemented in the brain without altering structural connectivity remains an open question in systems neuroscience. Although it is often overlooked in theories of whole-brain function, the neuromodulatory ascending arousal system is well-placed to mediate this role[4]. The arousal system is comprised of a range of nuclei spread across the brainstem and forebrain that send wide-reaching axons to the rest of the central nervous system[5]. At their target sites, arousal neurons release neuromodulatory neurotransmitters that shape and constrain a region's processing mode—altering their excitability and responsivity without necessarily causing them to fire an action potential[4,6]. As a result, subtle changes in the concentration of neuromodulatory chemicals can cause massive alterations in the dynamics of the target regions, leading to nonlinear effects on the coordinated patterns of activity that emerge from 'simple' neuronal circuits[4].

The ascending arousal system also contains substantial heterogeneity – unique cell populations project in diverse ways to the cerebral cortex and release distinct neurotransmitters. One key dichotomy is the distinction between adrenergic neuromodulation (predominantly via the locus coeruleus, LC), which promotes arousal and exploratory behaviour[7], and cholinergic neuromodulation (such as via the basal nucleus of Meynert, BNM), which is associated with attentional focus and vigilance[8]. These highly interconnected[9] structures both promote wakefulness and arousal[10,11], albeit via distinct topological projections to the cerebral cortex: the LC projects in a diffuse manner that crosses typical specialist boundaries, whereas the BNM projects in a more targeted, region-specific manner[12] (Fig. 1a). The two systems have also been linked with distinct and complimentary computational principles: the noradrenergic LC is presumed to modulate interactions between neurons (multiplicative gain; Fig. 1a, red)[13], whereas the cholinergic BNM is presumed to facilitate divisive normalization (response gain; Fig. 1a, green)[14]. Based on these anatomical and computational features, we have hypothesized that the interaction between these two neuromodulatory systems is crucial for mediating the dynamic, flexible balance between integration and segregation in the brain[15].

Another crucial feature of the ascending arousal system is that the number of neurons that project to the cerebral cortex is several orders of magnitude smaller than those that project back to the brainstem and forebrain[16–18]. Based on this feature, we further hypothesize that shifts in arousal are realized through a low-dimensional modulation of the ongoing neural activity

('brain state')[17]. Conceptually, low-dimensional neural dynamics can be depicted as evolving on a brain-state energy landscape[19], where the energy of a given state corresponds to the probability of occurrence, e.g. high-energy brain states have a low probability of occurrence (and v.v.). Brain states evolve along the energy landscape topography, much like a ball rolls under the influence of gravity down a valley and requires energy to traverse up a hill, this corresponds to an evolution towards an attractive or repulsive brain state, respectively. This technique can resolve what might otherwise be obscured states of attraction (and repulsion) in a multi-stable system and has been successfully applied to the dynamics of spiking neurons[20,21], blood oxygenation level-dependent (BOLD) functional magnetic resonance imaging (fMRI)[22,23] and magnetoencephalography (MEG)[24]. The approach offers several conceptual advances, but perhaps most importantly, it renders the otherwise daunting task of systems-level interpretation relatively intuitive.

In this manuscript we test these ideas by combining high-resolution resting-state fMRI data with analytic techniques from the study of complex systems. We show the low-dimensional landscape framework is not a mere analogy, as the topography of the energy landscape shares a one-to-one correspondence with the theorized role of the ascending arousal system through their effect on neural gain. We then extend our analysis to a cohort of experienced meditators and demonstrate changes in the cortical dynamical signatures following LC-mediated network dynamics were associated with internal shifts in conscious awareness. Together, these results present a view of brain organization that highlights the ascending arousal system's role in shaping both the dynamics of the cerebral cortex and conscious awareness.

## Results

To begin with, we extracted time series data from major subcortical hubs within the noradrenergic LC[9] (Fig. 1a, red; Supplementary Fig. 1) and cholinergic BNM[25] (Fig. 1a, green) systems from 59 healthy participants who had undergone high-resolution, 7 T resting-state fMRI (2 mm³ voxels; TR = 586 ms repetition time). Given the known spatiotemporal interactions between the ascending arousal system and fluctuations in cerebrospinal fluid, we first controlled for activity fluctuations in the nearby fourth ventricle, which contains no neural structures, but nonetheless can cause alterations in the BOLD signal over time. We next accounted for nearby grey-matter signals, by regressing the signal from the nearby pontine nuclei. Using the residuals from these regressions from the LC signal, $\tau_{LC}$ and the BMN signal, $\tau_{BNM}$, we focused on the difference between these signals ($\tau_{LC-BNM}$ and $\tau_{BNM-LC}$) and then identified time points associated with phasic bursts of LC activity (identified via peaks in the signals second derivative) that led to sustained adrenergic (vs. cholinergic) influence over evolving brain-state dynamics (and v.v. for phasic bursts of BNM; see 'Methods'). Importantly, the phasic mode of firing within the noradrenergic arousal system has been specifically linked to systemic influences that occur on time scales relevant to cognitive function[8,26]. Tracking the mean cortical BOLD response around these peaks identified a spatiotemporal travelling wave (Fig. 1b; velocity = 0.13 m s⁻¹) that propagated from frontal to sensory cortices and tracked closely with the known path of the dorsal noradrenergic bundle[9], albeit with a preserved island within the parietal operculum (Fig. 1b). A block-resampling null model was applied to ensure that the results were not due to spatial autocorrelation ($p < 0.05$; see 'Methods'). These results are inverted for BNM phasic activity (relative to LC) as $\tau_{BNM-LC} = -\tau_{LC-BNM}$, thus, BNM activity elicited a travelling wave that propagated from sensory to frontal cortices. Furthermore, these results confirm that coordinated

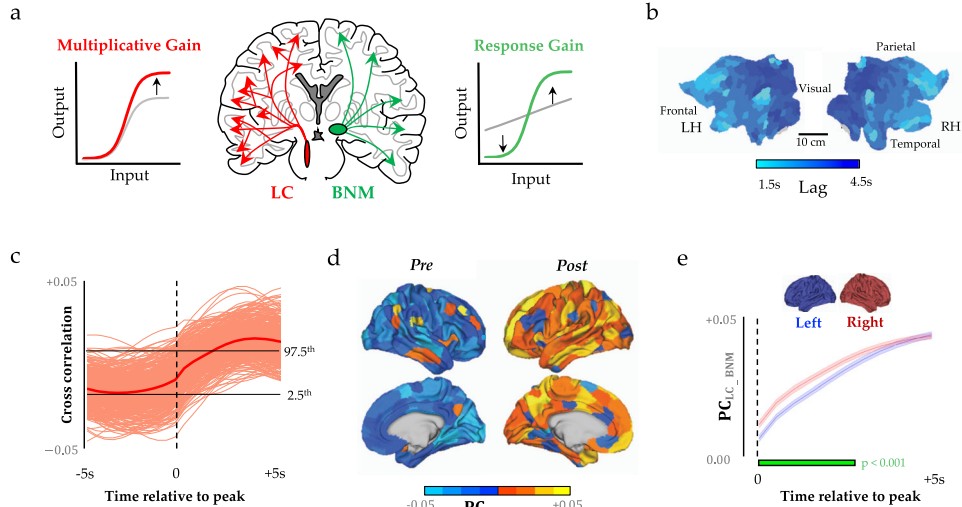

**Fig. 1 Sympathetic activity precedes network-level integration. a** Regional time series were extracted from the subcortical locus coeruleus (red), which is thought to alter multiplicative gain, and the basal nucleus of Meynert (green), which is thought to alter response gain, and they were compared to cortical BOLD signal and topological signatures during the resting state. **b** We observed an anterior-to-posterior travelling wave (velocity ~ 0.13 m s⁻¹) following peaks in $\tau_{LC\text{-}BNM}$, which are shown on both the left (LH) and right (RH) hemispheres of a cortical flat map (and v.v. following peaks in $\tau_{BNM\text{-}LC}$). **c** Lagged cross-correlation between $\tau_{LC\text{-}BNM}$ and PC for each parcel (faint line) and mean PC (solid line); dotted line depicts the zero-lag correlation, and the black lines depict the upper (lower) bounds of the block-resampled null model 95% CI. **d** Mean cortical participation coefficient (PC) preceding (left) and following (right) the zero-lagged $\tau_{LC\text{-}BNM}$ value, only the right hemisphere is shown, which is mirrored for Post. **e** Mean participation coefficient following peak $\tau_{LC\text{-}BNM}$ was higher in the right (red) vs. the left (blue) hemisphere (error bars represent mean± SEM, $n = 200$ parcels within a hemisphere, $p < 0.001$ green bar one-sided permutation test 5000 random permutations).

macro-scale activity patterns align to fluctuations in activity within the ascending arousal system of the brainstem[27].

**Time-varying network topology.** Based on previous empirical[28], modelling[29] and theoretical[15] work, we predicted that phasic bursts in $\tau_{LC\text{-}BNM}$ would facilitate network-level integration by modulating increased neural gain among regions distributed across the cerebral cortex. As predicted, we observed a strong positive correlation between $\tau_{LC\text{-}BNM}$ and network-level integration ($p < 0.05$, block-resampling null model; Fig. 1c) across the brain (Fig. 1d). An increase in phasic activity within the LC (relative to the BNM) preceded an increase in the mean level of integration (Fig. 1d) within the cerebral cortex that was dominated by the frontoparietal cortices (Supplementary Fig. 2; parcellated according to the 17 resting-state networks identified in[30]). Interestingly, this global integration was opposed by a relative topological segregation of limbic, visual and motor cortices (Supplementary Fig. 2). This increase in the synchronization of the frontoparietal cortices following an increase in sensory–limbic coordination and LC activity may reflect arousal-enhanced processing of sensory stimuli[31,32]. Furthermore, regional integration occurred earlier in the right vs. the left hemisphere ($p < 0.001$; Fig. 1e), which is consistent with the known anatomical bias of the LC system[33,34]. Together, these findings provide robust evidence for the hypothesis that the balance between ascending noradrenergic and cholinergic tone facilitates a transition towards topological integration across the frontoparietal network of the brain[15].

**Neuromodulation of the energy landscape.** The results of our initial analysis demonstrate that coordinated distributed BOLD activity in the cortex aligns with changes in small groups of neuromodulatory cells BOLD activity in the brainstem and forebrain, which in turn are proposed to constrain brain dynamics onto a low-dimensional energy landscape (Fig. 2a). The effects of noradrenaline and acetylcholine can also be easily viewed through this lens: by integrating the brain, noradrenaline

should flatten the energy landscape (Fig. 2a, red) facilitating otherwise unlikely brain-state transitions, whereas the segregative nature of acetylcholine should act to deepen energy valleys (Fig. 2a, green) decreasing the likelihood of a brain-state transition. In previous work, we have shown a correspondence between low-dimensional brain-state dynamics across multiple cognitive tasks and the heterogenous expression of metabotropic neuro-modulatory receptors[17]. This implies that neuromodulators act similar to catalysts in chemical reactions, which lower (or raise) the activation energy ($E_A$) required to transform chemicals from one steady-state (or energy well) to another (Fig. 2b). In the context of the interconnected, heterarchical networks that comprise the cerebral cortex, this would have the effect of flattening (or deepening) the energy landscape, promoting variable (or rigid) brain states[35] (Fig. 2a).

To elucidate the role of phasic activity from the neuromodulatory system in modifying the energy landscape, we first estimated the energy of BOLD signal transitions across the cerebral cortex. Importantly, the term 'energy' here is used in reference to its definition in statistical physics and hence does not represent the biological use of the term, which instead stands for the metabolic energy used by the brain to maintain or change neural activity. Specifically, we define the energy landscape, $E$, as the natural logarithm of the inverse probability of observing a given BOLD mean-squared displacement (MSD) at a given time-lag $t$, P(MSD, $t$), calculated as $E = \ln\left(\frac{1}{P(MSD,t)}\right)$, where $MSD_{t,t_0} = \left\langle \left| x_{t_0+t} - x_{t_0} \right|^2 \right\rangle_r$ is the MSD of BOLD signal, $x_t = [x_{1,t}, x_{2,t}, \ldots, x_{r,t}]$ across $r$ voxels and $t$ is the number of time-lags of size TR from a reference time point $t_0$[20]. The probability of a BOLD signal transition, P(MSD, $t$), was estimated from the sampled $MSD_{t,t_0}$, calculated using a Gaussian kernel density estimation $P(MSD, t) = \frac{1}{4n}\sum_{i=1}^{n} K\left(\frac{MSD_{t,i}}{4}\right)$, where $K(u) = \frac{1}{2\sqrt{\pi}}e^{-\frac{1}{2}u^2}$

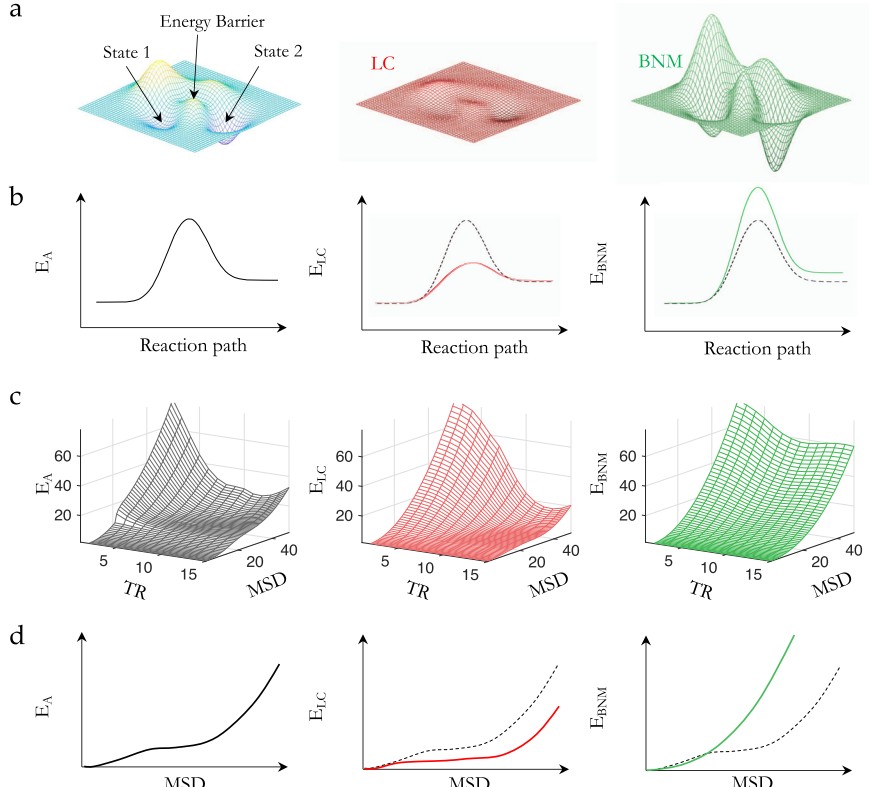

**Fig. 2 LC and BNM mediated shifts in the brain-state energy landscape. a** An example energy landscape, which defines the energy required to move between different brain states: by increasing multiplicative gain, the LC should flatten the energy landscape (red); by increasing response gain, the BNM should accentuate the energy wells (green). **b** The topography of the energy landscape can be conceptualized as similar to the activation energy ($E_A$) that must be overcome in order to convert one chemical to another. **c** Empirical BOLD energy landscape as a function of mean-squared displacement (MSD) and TR of the baseline activity ($E_A$, black) and after phasic bursts in LC ($E_{LC}$, red) and BNM ($E_{BNM}$, green). **d** Empirical activation energy as a function of MSD averaged across lags $t = [10 : 15]$ TR during base baseline activity ($E_A$, Left) and following phasic bursts in LC ($E_{LC}$, red) and BNM ($E_{BNM}$, green). Relative to the baseline energy landscape, phasic bursts in LC lead to a flattening or reduction of the energy landscape, whereas peaks in BNM lead to an accentuation of the well by raising the energy landscape.

(see 'Methods'). Our analysis is consistent with the statistical mechanics interpretation that the energy of a given state, $E_\sigma$, and its probability are related $P_\sigma = \frac{1}{Z} e^{\frac{E_\sigma}{T}}$, where $Z$ is the normalization function and $T$ is a scaling factor equivalent to temperature in thermodynamics, where we set $T = 1$ and $Z = 1$, thus, $P_\sigma = e^{-E_\sigma} \rightarrow E_\sigma = \ln\left(\frac{1}{P_\sigma}\right)$[20]. In this framework, a highly probable relative change in BOLD (as quantified by the MSD) corresponds to a relatively low energy transition (i.e. low $E_A$), whereas an infrequently visited state will require the most energy (i.e. high $E_A$).

By treating energy as inversely proportional to the probability of brain state occurrence, our approach resembles other studies that have been applied to spiking dynamics of neuronal populations[20,21], BOLD fMRI[22,23], MEG[24] and natural scene[36]. These studies binarized continuous signals to reduce the brain-state space (to $2^r$ states), however, this approach requires artificial thresholding, which can be problematic in continuously recorded data. In contrast, our approach reduces the dimensionality by analysing the likelihood of a change in BOLD activity (i.e. the MSD), and thus retains the continuous nature of the underlying signal without the need for thresholding. Furthermore, this approach overcomes a major limitation inherent to previous energy landscape studies that require a large sample size to sufficiently sample the brain-state space.

With this in mind, we turned our attention to the relationship between the ascending arousal system dynamics and the MSD energy landscape. To test the hypothesis that the

neuromodulatory system alters the topography of the energy landscape, we calculated BOLD MSD energetics following phasic bursts of both LC relative to BNM ($\tau_{LC-BNM}$), $E_{LC}$ and BNM relative to LC ($\tau_{BNM-LC}$), $E_{BNM}$, i.e. $t_0$ was the onset of a phasic burst, and we contrasted these with brain evolutions outside of phasic bursts in LC and BNM, termed the baseline energy landscape $E_A$. We identified phasic bursts as peaks in the second derivative of the arousal BOLD signals $\tau_{LC-BNM}$ and $\tau_{BNM-LC}$ that lead to a sustained increase in BOLD activity for each individual (see 'Methods') and using these criteria, we identified 148 $\tau_{LC-BNM}$ time points and 130 $\tau_{BNM-LC}$ time points.

The energy landscapes for these three states are defined by the energy for a given mean change in BOLD activity (i.e. MSD) at a given temporal displacement (i.e. TR). Figure 2c demonstrates the baseline energy landscape (Fig. 2c, black), and the change to the energy landscape following phasic bursts in the LC (Fig. 2c, red) or BNM (Fig. 2c, green). We found the largest change occurs around 10–15 TR (~6–9 s) following a phasic burst that typically corresponds to a peak in the LC or BNM BOLD signal. At this ~6–9 s temporal delay, we see direct evidence that a phasic burst of LC flattened the energy landscape (decreased the energy relative to baseline Fig. 2c, red inset), thus making previously unlikely state trajectories far more probable (Fig. 2d, red), whereas a phasic burst of BNM activity (increased energy relative to baseline Fig. 2c, green inset) caused the energy landscape to be elevated, thus promoting local trajectories, and making large state deviations unlikely (Fig. 2d, green). These patterns are analogous to modulating a physical landscape in which towns sit within

valleys separated by impassable mountains—when BNM is high, the towns remain isolated, whereas when LC is high, transitions between towns can be easily realized.

We next asked whether LC and BNM combined synergistically to alter the energy landscape. To achieve this, we isolated simultaneous phasic peaks in both LC and BNM ($\tau_{LC+BNM}$). We found that the LC + BNM energy landscape differed from either independent LC or BNM activation, shifting the brain state into divergent regimes than could be explained by the HRF. By comparing the MSD energy topography for a given TR slice we found that the landscape switched from an anti- to de-correlation with the HRF (Supplementary Fig. 3a). In other words, the cooperative behaviour between the noradrenergic and cholinergic systems allowed the brain to reach unique BOLD MSDs that neither could facilitate individually. To examine how simultaneous LC + BNM activity altered the energy landscape, we compared the energy relative to the two individual landscapes. As demonstrated in Supplementary Fig. S3b, the energy landscape following phasic bursts of LC + BNM differed in magnitude from that expected from a linear superposition of the LC and BNM energy landscape—i.e. LC + BNM ≠ (LC) + (BNM). Furthermore, to explore the dominance of either LC or BNM in this signal, we minimized the relationship LC + BNM = $\alpha$LC + $\beta$BNM (conditional upon $\alpha$ and $\beta$ being positive constants) and found that $\alpha = 0.16$ and $\beta = 0.84$ gave the best match to the LC + BNM energy landscape. That is, the BNM dynamics dominates the simultaneous LC + BNM energy landscape, which is consistent with the unidirectional synaptic projections from the LC that synapse upon the BNM on their way through to the cortex[12], and suggests that phasic LC + BNM bursts may be initiated by the LC in order to elicit a cascade of BNM activity.

**Conscious awareness of shifts in BOLD state.** Interpreting the relationship between neuroimaging data and conscious awareness is notoriously challenging. For instance, it is currently not possible to directly determine the contents of self-directed thought without intervening, and thus altering, the contents of consciousness[37]. Although we cannot determine the contents of consciousness directly, we can use task designs to modulate the state of consciousness. To this end, we leveraged data from a group of 14 expert meditators who were asked to meditate during an fMRI scanning session[38], and to press a button when they noticed that their focus had drifted from their breath (Fig. 3a). At this point, there is a mismatch between expectation and conscious awareness, which is an internal state that has been previously linked to the activation of the noradrenergic system, both in theoretical[39,40] and computational[41] work. Based on these studies, we predicted that the switch in internal conscious awareness would be facilitated by increases in LC-mediated integration and subsequent reconfiguration of low-dimensional brain states. Analysing time-resolved network data with a finite impulse response model, we observed a peak in LC activity (Fig. 3b), TR-to-TR BOLD MSD (Fig. 3c) and elevated network-level integration (Fig. 3d) surrounding the change in conscious awareness (all $p_{PERM} < 0.05$; 95% CI of null distribution calculated from 5000 permutation tests). These results confirm that the LC mediates energy landscape reconfigurations and that these changes modulate internal states of conscious awareness.

**Discussion**
Our results provide evidence for arousal-mediated macroscopic network and energy landscape reconfiguration which track with moment-to-moment alterations in conscious awareness. By tracking fluctuations in BOLD signal within the noradrenergic LC and the cholinergic BNM, we were able to demonstrate

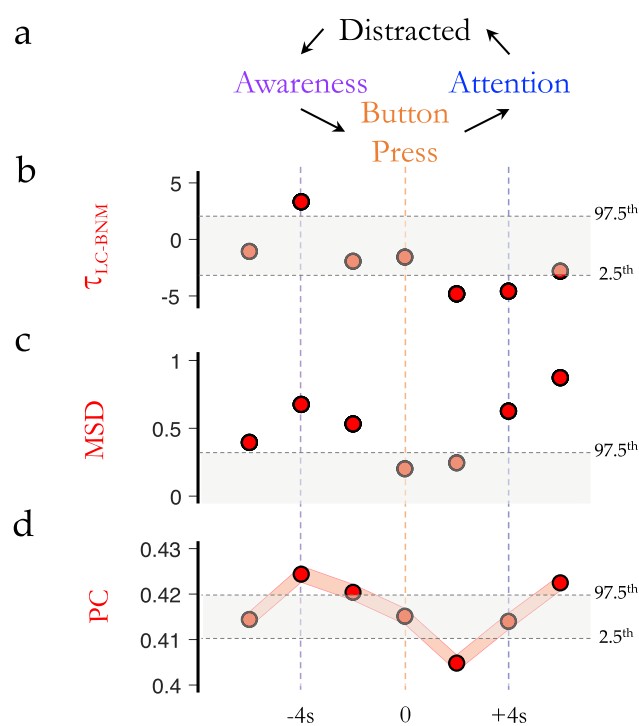

**Fig. 3 Awareness of intrinsic state changes. a** Participants performing breath-awareness meditation (focus; blue) were trained to respond with a button press (orange) when they became aware (purple) that they had become distracted (i.e. their attention had wandered from their breath) and to then re-focus their attention (blue) on their breath. **b** We observed a peak in $\tau_{LC-BNM}$ (red; LC relative to BNM BOLD activity) ~4 s before the button press, which then returned to low levels in the 2–4 s following the button press. **c** The mean-squared displacement (MSD; dark orange) of TR-to-TR BOLD signal was increased above null values around the peak in $\tau_{LC-BNM}$, as well as following the re-establishment of attentional focus (in (**b**, **c**) grey shading depicts 95% CI of block-resampled null distribution). **d** We observed a peak in mean participation coefficient (PC) ~4 s (2 TRs) prior to the button press during the task. **a–d** Grey shading depicts mean-centred 97.5th and 2.5th percentile of block-resampled null distribution 5000 permutations, i.e. outside grey shading indicates a value different than null ($p < 0.05$); and (**d**) red shading represents mean ± SEM across ($n = 400$ parcels). Source data are provided as a Source Data file.

fundamental ways in which the low-dimensional, dynamic and topological signature of cortical dynamics are related to changes within the ascending arousal system. Furthermore, we demonstrated a link between these dynamic reconfigurations and alterations in conscious awareness in a cohort of experienced meditators. In this way, our results provide a systems-level perspective on the distributed dynamics of the human brain.

There is growing evidence that macro-scale neural dynamics in the brain are well described by relatively low-dimensional models[17,18,42–44], however, the biological constraints that impose these features on the brain remain poorly understood. Due to the low number of cells in the arousal system and their broad projections to the rest of the brain, we theorized that neuromodulatory regions are well-placed to shape and constrain the vast number of neurons in the cerebral cortex into low-dimensional dynamical manifolds. Our results support this prediction by showing that patterns of activity in key regions within the brainstem and forebrain relate to fundamental alterations in a dynamically evolving energy landscape. In other words, neural state space trajectories are a powerful framework that extends

beyond that of mere analogy, and the ascending arousal system is well-placed to mediate deformations in the energy landscape.

Much in the same way that there are many different reference frames for navigation—e.g. egocentric (i.e. straight, left, right directions), which is independent of the environment; and allocentric (i.e. following compass directions and visual cues), which is dependent on the environment—we can interrogate energy landscapes using different vantage points on BOLD dynamics. Our displacement framework is consistent with an egocentric (or first-person) frame of reference, wherein MSD is used to track BOLD trajectories from an initial state which maps out the topology of the energy landscape (i.e. a BOLD MSD implies a BOLD trajectory). Nonetheless, the method does not distinguish between two different neural trajectories that possess the same MSD. In comparison, other methods have evaluated the energy landscape for a given pre-defined state estimated from thresholded BOLD time series[22,23] a framework consistent with an allocentric (or third-person) reference frame, i.e. state-activity dependent. The allocentric framework has the advantage of calculating energy for a given state, however, it also requires substantial exploration of the state-space—which is typically unfeasible—or the need to resort to severe coarse-graining (such as the binarization of BOLD activity) which further diminishes interpretability. Furthermore, the allocentric view does not provide insights into the transitions between each energy state, whereas this information is inherent to the egocentric reference. Along these lines, we found that the egocentric reference frame clearly demonstrated the flattening and deepening of the energy landscape, providing indirect evidence that the ascending arousal system is well set-up to control brain-state dynamics 'egocentrically'. Nevertheless, given improvements in recording length and new techniques to probe the brains dynamical landscape, we expect that the field will ultimately discover even more optimal mappings between neurobiology and low-dimensional brain-state dynamics.

The results of our state-space analysis have important implications for the biological mechanisms underlying cognition. For instance, the concept of LC-mediated energy landscape flattening is reminiscent of the α1 receptor-mediated notion of a network reset[39]. By increasing response gain (Fig. 1a) through the modulation of second-messenger cascades[4], noradrenaline released by the LC would augment inter-regional coordination[29]. Importantly, this capacity could confer adaptive benefits across a spectrum, potentially facilitating the formation of flexible coalitions in precise cognitive contexts[45], while also forcing a broader landscape flattening (i.e. a reset) in the context of large, unexpected changes[26,39]. Similarly, the idea that phasic cholinergic bursts accentuate energy wells is consistent with the idea that the cholinergic system instantiates divisive normalization within the cerebral cortex[14]. Numerous cognitive neuroscience studies have shown that heightened acetylcholine levels correspond to improvements in attentional precision[8]. By accentuating energy wells, acetylcholine from the BNM could ensure that the brain remains within a particular state and is hence not diluted by other (potentially distracting) brain states. Determining the specific rules that govern the links between the neuromodulation of the energy landscape and cognitive function[46–48] is of paramount importance, particularly given the highly integrated and degenerate nature of the ascending arousal system[49].

Our results also provide a systems-level perspective on an emerging corpus of work that details the microscopic circuit level mechanisms responsible for conscious phenomena[50]. In particular, a number of recent studies have highlighted the intersection between the axonal projections of the ascending arousal system and pyramidal cell dendrites in the supragranular regions of the cerebral cortex as a key site for mediating conscious awareness.

For instance, optogenetic blockage of the connections between the cell bodies and dendrites of thick-tufted layer V pyramidal cells in the sensory cortex causally modulated conscious arousal in mice[51]. Other works have shown that both the noradrenergic[52] and cholinergic[53] systems alter this mechanism, albeit in distinct ways: noradrenaline would promote burst firing due to the α2a receptor-mediated closure of $Ih$ HCN leak-channels[52], whereas the cholinergic system instead prolongs the time-scale of firing via M1 cholinergic receptor activation on pyramidal cell dendrites[53]. In this way, coordinated activity in the ascending arousal system can mediate alterations in microcircuit processing that ultimately manifest as alterations in macroscopic brain network dynamics.

The vascular nature of the T2* fMRI signal is such that it is impossible to rule out the role of haemodynamics in the results we obtained in our analysis. Indeed, there is evidence that noradrenaline causes a targeted hyperaemia through the augmentation of G-protein-coupled receptors on vascular smooth muscle cells[54,55]. However, it is also clear that the haemodynamics and massed neural action in the cerebral cortex are inextricably linked[56,57]. In addition, there is evidence that stimulation of the LC leads to the high-frequency, low-amplitude electrophysiological activity patterns characteristic of the awake state[10]. Together, these results argue that the LC mediates a combination of haemodynamic and neural responses that facilitate integrative neural network interactions and subsequently mediate alterations in conscious awareness.

In this manuscript, we have argued that the ascending arousal system provides crucial constraints over normal brain function, however, there are numerous examples wherein pathology within the ascending arousal system leads to systemic impairments in cognition. In addition to disorders of consciousness[58], dementia syndromes are also crucially related to dysfunction within the ascending arousal system. For instance, Alzheimer's disease has been linked to tau pathology within the BNM[25], however, individuals with Alzheimer's disease also often have pathological involvement of the LC as well[59]. Similarly, individuals with Parkinson's disease often have extra-dopaminergic pathology in the LC[60], as well as in the cholinergic tegmentum[61]. Given the pathological processes at play in these disorders, we expect that other neuromodulatory systems will also be impaired, and in turn affect the macroscopic dynamics of the system in ways that remain to be elucidated.

In conclusion, we leveraged a high-resolution 7 T resting-state fMRI dataset to test the hypothesis that activity within the ascending arousal system shapes and constrains patterns of systems-level network reconfiguration. Our results support specific predictions from a recent hypothetical framework[15], and further delineate how the autonomic nervous system shapes and constrains ongoing, low-dimensional brain-state dynamics in the central nervous system in a manner that supports changes in conscious awareness.

## Methods

**7T resting-state fMRI**. These data were originally described in Hearne et al. (2017) where full experimental details can be found[62], we selectively analysed the first resting-state recordings obtained from this dataset. Briefly, we outline the data collection. Sixty-five healthy, right-handed adult participants (mean, 23.35 years; s.d., 3.6 years; range 18–33 years; 28 females) were recruited, of whom 59 were included in the final analysis (four participants were excluded due to MR scanning issues, one participant was excluded due to an unforeseen brain structure abnormality, and one was excluded due to inconsistent BOLD dynamics following global-signal regression). Participants provided informed written consent to participate in the study. The research was approved by The University of Queensland Human Research Ethics Committee. 10 min of whole-brain 7 T resting-state fMRI echo planar images were acquired using a multiband sequence (acceleration factor = 5; 2 mm³ voxels; 586 ms TR; 23 ms TE; 40° flip angle; 208 mm FOV;

55 slices). Structural images were also collected to assist functional data pre-processing (MP2RAGE sequence: 0.75 mm$^3$ voxels; 4300 ms TR; 3.44 ms TE; 256 slices).

Imaging data were pre-processed using an adapted version of MATLAB (MathWorks R2020a). DICOM images were first converted to NIfTI format and realigned. T1 images were reoriented, skull-stripped (FSL BET) and co-registered to the NIfTI functional images using statistical parametric mapping functions. Segmentation and the DARTEL algorithm were used to improve the estimation of non-neural signal in subject space and the spatial normalization. From each grey-matter voxel, the following signals were regressed: linear trends, signals from the six head-motion parameters (three translation, three rotation) and their temporal derivatives, white matter and CSF (estimated from single-subject masks of white matter and CSF). The aCompCor method[63] was used to regress out residual signal unrelated to neural activity (i.e. five principal components derived from noise regions-of-interest in which the time series data were unlikely to be modulated by neural activity). Participants with head displacement >3 mm in >5% of volumes were excluded. A temporal band pass filter ($0.01 < f < 0.15$ Hz) was applied to the data.

**Brain parcellation**. Following pre-processing, the mean time series was extracted from 400 pre-defined cortical parcels using the Schaefer atlas[64]. Probabilistic anatomical atlases were used to define the location of the noradrenergic LC[65] and the cholinergic BNM (Ch4 cell group)[66]. The mean signal intensity from each region was extracted and then used for subsequent analyses. To ensure that the BOLD data were reflective of neuronal signals, we statistically compared LC and BNM time series with a number of potential nuisance signals from: (i) the cerebrospinal fluid; (ii) the cortical white matter; (iii) mean frame-wise displacement; and (iv) a 2 mm$^3$ sphere in the fourth ventricle (centred at MNI coordinates: 0 −45 −30)[67]. All signals were unrelated to LC and BNM activity ($|r| < 0.05$ in each case), however, given the spatial proximity of the LC to the fourth ventricle, we opted to use a linear regression to residualize the signal from the fourth ventricle. To ensure that BOLD signals from nearby grey-matter structures were not influencing the LC time series, we extracted the mean activity of the LC mask after shifting the mask anteriorly such that it overlapped with an area of the pons that harbours the nuclei (i.e. +8 mm in the $Y$ direction). In the same manner in which we previously regressed the dynamics of the fourth ventricle, we regressed the activity of this non-LC pontine region, and then re-analysed our data. Each of the results was statistically identical following this approach, providing confidence that the original conclusions were not biased by a lack of regional specificity. However, as the LC is surrounded by various other arousal controlling nuclei, the signal likely contains BOLD activity of adjacent nuclei. We further expected this issue to be more significant with the reduced spatial resolution in the 3 T recording, nevertheless, the similarity in the findings between the 3 T and the 7 T analysis provides confidence of our claims.

**Phasic increases in neuromodulatory BOLD signal**. To identify phasic increases in neuromodulatory BOLD signal, we calculated the second derivative (i.e. the acceleration) of the LC and BNM time series, and then identified points in time that fulfilled three criteria: (1) value greater than or equal to 2 s.d. above the mean acceleration; (2) value of the original time series, i.e. LC or BNM, was greater than or equal to 2 s.d. above the mean of the time series within the following 10 TRs (i.e. 5.8 s); and (3) the time point was not present within the first or last 20 TRs of an individual subjects' trial (so as to avoid potential boundary effects). Using these criteria, we identified 148 $\tau_{\text{LC-BNM}}$ time points (mean 2.5 per individual with a range between [0 5]), 130 $\tau_{\text{BNM-LC}}$ time points (mean 2.2 per individual with a range between [0 7]) and 316 $\tau_{\text{LC+BNM}}$ time points (mean 5.4 per individual with a range between [2 10]) across all 59 subjects. To ensure that the choice of 2 s.d. threshold was reflective of the underlying dynamics, we altered this threshold between 0.5 and 2.5 s.d. and found robustly similar patterns. For subsequent analyses, we identified time points in the 21 TR window surrounding these peaks, and then used these to conduct statistical comparisons of the low-dimensional, complex network signature of brain network dynamics as a function of phasic ascending arousal system activity. Each of these patterns was confirmed using a lag-based cross-correlation analysis, which demonstrated similar phenomena to those that we present in the manuscript.

To monitor the propagation of cortical signals with respect to $\tau_{\text{LC-BNM}}$, $\tau_{\text{BNM-LC}}$ and $\tau_{\text{LC+BNM}}$, we extracted the time-to-peak of the cross-correlation between these signals and each of the 400 cortical parcels within the 10 TR (i.e. 5.8 s) windows following each identified phasic peak. These patterns were mapped onto the cortex (Fig. 1b) for visualization and clearly demonstrated an anterior-to-posterior direction for the wave. We then used the volumetric MNI coordinates of the Schaefer parcellation scheme to calculate the average velocity of the travelling wave (~0.13 m s$^{-1}$).

In order to obtain an appropriate null model against which to compare our data, we identified 5000 random time points within the concatenated dataset that did not substantially overlap with the already identified $\tau_{\text{LC-BNM}}$, $\tau_{\text{BNM-LC}}$ and $\tau_{\text{LC+BNM}}$ time series, and used these to populate a null distribution[68]. Outcome measures were deemed significant if they were more extreme than the 95th (or 5th) percentile of the null distribution. Crucially, this ensured that our data could not be explained by the characteristic spatial and temporal autocorrelation present in BOLD time series data.

**Time-resolved functional connectivity**. To estimate functional connectivity between the 400 regions-of-interest, we used the multiplication of temporal derivatives (MTD) technique. Briefly, MTD is computed by calculating the point-wise product of temporal derivative of pair-wise time series. The resultant score is then averaged over a temporal window, w (a window length of 20 TRs was used in this study, though results were consistent for $w = 10$–$50$ TRs).

**Modularity maximization**. The Louvain modularity algorithm from the Brain Connectivity Toolbox (BCT)[69] was used on the neural network edge weights to estimate community structure. The Louvain algorithm iteratively maximizes the modularity statistic, $Q$, for different community assignments until the maximum possible score of $Q$ has been obtained:

$$Q_T = \frac{1}{v^+} \sum_{ij} \left( w_{ij}^+ - e_{ij}^+ \right) \delta_{M_i M_j} - \frac{1}{v^+ + v^-} \sum_{ij} \left( w_{ij}^- - e_{ij}^- \right) \delta_{M_i M_j},$$

where $v$ is the total weight of the network (sum of all negative and positive connections), $w_{ij}$ is the weighted and signed connection between regions $i$ and $j$, $e_{ij}$ is the strength of a connection divided by the total weight of the network, and $\delta_{M_i M_j}$ is set to 1 when regions are in the same community and 0 otherwise. '+' and '−' superscripts denote all positive and negative connections, respectively. The modularity of a given network is therefore a quantification of the extent to which the network may be subdivided into communities with stronger within-module than between-module connections.

For each epoch, we assessed the community assignment for each region 500 times and a consensus partition was identified using a fine-tuning algorithm from the Brain Connectivity Toolbox (BCT; http://www.brain-connectivity-toolbox.net/ ). We calculated all graph theoretical measures on un-thresholded, weighted and signed connectivity matrices. The stability of the γ parameter was estimated by iteratively calculating the modularity across a range of γ values (0.5–2.5; mean Pearson's $r = 0.859 \pm 0.01$) on the time-averaged connectivity matrix for each subject—across iterations and subjects, a γ value of 1.0 was found to be the least variable, and hence was used for the resultant topological analyses.

**Participation coefficient**. The participation coefficient, PC, quantifies the extent to which a region connects across all modules (i.e. between-module strength) and has previously been used to successfully characterize hubs within brain networks. The PC for each region was calculated within each temporal window as,

$$\text{PC} = 1 - \sum_{s=1}^{n_M} \left( \frac{\kappa_{isT}}{\kappa_{iT}} \right)^2$$

where $k_{isT}$ is the strength of the positive connections of region $i$ to regions in module $s$ at time $T$, and $k_{iT}$ is the sum of strengths of all positive connections of region $i$ at time $T$. Negative connections were discarded prior to calculation. The participation coefficient of a region is therefore close to 1 if its connections are uniformly distributed among all the modules and 0 if all its links are within its own module.

**Brain-state displacement and the energy landscape**. To quantify the change in BOLD activity following phasic bursts of neuromodulation we calculated the BOLD MSD. The MSD is a measure of the deviation in BOLD activity, $\boldsymbol{x}_t = [x_{1,t}, x_{2,t}, \ldots , x_{r,t}]$ for $r$ parcels, with respect to the activity at the phasic onset, $t_0$. The MSD is calculated as the average change of each voxel

$$\text{MSD}_{t,t_0} = \left\langle \left| \boldsymbol{x}_{t_0+t} - \boldsymbol{x}_{t_0} \right|^2 \right\rangle_r,$$

where $<>_r$ is the mean across $r$ parcels, and it is calculated for different $t_0$, where $t_0$ are the onset of a subcortical phasic burst, across $t$ TRs. We are interested in the probability, $P(\text{MSD}, t)$ that we will observe a given displacement in BOLD at a given time-lag $t$. We estimated the probability distribution function from $n$ $\text{MSD}_{t,t_0}$ samplings—e.g. the identified $n$ phasic bursts of subcortical structures (as above)— using a Gaussian kernel density estimation $P(\text{MSD}, t) = \frac{1}{4n} \sum_{i=1}^{n} K\left( \frac{\text{MSD}_{t,i}}{4} \right)$, where $K(u) = \frac{1}{2\sqrt{\pi}} e^{-\frac{1}{4}u^2}$ and we display the results for $t$ between 1 and 15 TR and MSD between 0 and 50. As is typical in statistical mechanics the energy of a given state, $E_\sigma$, and its probability are related $P(\sigma) = \frac{1}{Z} e^{-\frac{E_\sigma}{T}}$, where $Z$ is the normalization function and $T$ is a scaling factor equivalent to temperature in thermodynamics[20]. In our analysis $\Sigma_\sigma P_\sigma = 1 \rightarrow Z = 1$ by construction and we can set $T = 1$ for the observed data. Thus, the energy of each BOLD MSD at a given time-lag $t$, $E$, is then equal to the natural logarithm of the inverse probability, $P(\text{MSD}, t)$, of its occurrence

$$E = \ln \left( \frac{1}{P(\text{MSD}, t)} \right).$$

**Meditation dataset**. These data were originally described in Hasenkamp et al. (2012) where full experimental details can be found[38]. Briefly, we outline the data collection. Fourteen healthy right-handed non-smoking meditation practitioners (11 female; age 28–66 years) underwent Siemens 3 T MRI scanning (T1: TR = 2600 ms, TE = 3.9 ms, TI = 900 ms, FOV = 24 cm, 256 × 256 matrix, voxel dimensions = 1 × 1 × 1 mm$^3$; T2*: weighted gradient-echo pulse sequence, TR = 1500 ms, TE = 30 ms, flip angle = 90°, FOV = 192 cm, 64 × 64 matrix, voxel dimensions = 3 × 3 × 4 mm$^3$). All participants signed a consent form approved by the Institutional Review Board at Emory University and the Atlanta Veterans Affairs Research and Development Committee as an indication of informed consent. Participants were asked to meditate for 20 min in the MRI scanner by maintaining focused attention on the breath and keeping the eyes closed. They were instructed to press a button whenever they realized their mind had wandered away from the breath, and then return their focus to the breath. The epoch of time immediately prior to the button press was thus the moment in time in which each individual recognized that their focus had deviated from their breath. This information was used to construct a finite impulses response model that mapped the five TRs prior-to and following each button press. We then modelled LC > BNM activity, low-dimensional dynamics and network topology around this epoch to construct a model of state-space reconfiguration as a function of intrinsic conscious awareness. Non-parametric, block-resampling null distributions were utilized for statistical testing ($p < 0.05$).

**Reporting summary**. Further information on research design is available in the Nature Research Reporting Summary linked to this article.

## Data availability
The resting-state BOLD cortical activity, subcortical LC and BNM activity, network statistics, and LC and BNM masks in MNI space have been deposited in a Zenodo database (https://doi.org/10.5281/zenodo.5315132). The source raw resting-state BOLD data that support the findings of this study were obtained from (Hearne et al., 2017)[62] and they are available from (http://data.qld.edu.au/public/Q1361/). The source raw meditation dataset was obtained from (Hasenkamp et al., 2012)[38], access can be obtained from the authors upon reasonable request. Source data are provided with this paper.

## Code availability
All the code required to conduct the analysis can be found on GitHub at (github.com/Bmunn/BSI; https://doi.org/10.5281/zenodo.5315765).

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

## Author contributions

J.S. conceived, funded, directed the project and curated the data. B.M., E.M. and J.S. conducted the analysis. B.M. and J.S. wrote the original draft. B.M., E.M., G.W. and J.S. reviewed and edited the manuscript.

## Competing interests

The authors declare no competing interests.
