## [Peer Review File. · Nature Communications]

The ascending arousal system shapes neural dynamics to mediate awareness of cognitive statesREVIEWER COMMENTS

Reviewer #1 (Remarks to the Author):

This interesting paper investigates the link between subcortical arousal systems and cortical state dynamics. Munn et al. aimed to determine how the ascending arousal system, specifically the LC and BNM nuclei, alters cortical gain and attractor dynamics using high-field fMRI. They find that phasic activity in specific subcortical regions is linked to the spatiotemporal dynamics of large-scale cortical networks. Their method to model energy dynamics provides a straightforward approach to studying attractor dynamics, and they provide interesting evidence to support their specific hypotheses about how ascending arousal system nuclei alter the brain's ability to change states. The approach and results provide a novel perspective on brainwide dynamics, and overall, the analyses provide useful information, although there are some issues with some of the methodological aspects.

Major comments:

- The study centers on interpretation of LC signals, but uses 2mm voxels. Since LC is only 1-2 mm in width, this suggests that other structures were present in the ROI. While supraresolution voxel sizes can be used successfully in certain structures with shapes that allow for averaging out the effects of the neighbouring structures, this is not the case for the LC. The paper should either show that similar dynamics are not obtained when studying signals from immediately adjacent voxels in those nearby structures, or should acknowledge/discuss that the measured signals reflect activity from LC and its neighbouring structures, and not overstate the specificity of the imaging. It would also be helpful to show an overlay of selected functional voxels on the anatomical localization of LC, describe the probability threshold used in the ROI definition, and report the number of voxels in the ROI.
- Due in part to this issue, some of the conclusions seem overstated. E.g., "The results of our initial analysis confirm that distributed activity in the brain is modulated by small groups of neuromodulatory cells in the brainstem and forebrain" after Fig. 1. I agree that the results are interesting, but this result does not show that neuromodulatory balance causally controls brain activity, particularly since LC is tiny and the LC voxel surely contained other structures as well – rather, this is already known to be true from animal studies, and this paper provides an interesting new analysis of that distributed activity. Slightly more cautious statements could be used throughout the results and discussion.
- There are no statistics for the lags across cortical regions, and it would be good to show whether at least earliest and latest are significantly different.

- Further explanation of the baseline dynamics would be helpful. Why are the energy landscapes in Figure 2C control shaped the way it is? Why does it decrease around 10? This would make the energy landscapes of LC-BNM and BNM-LC more meaningful.

- Fig 1C y-axis label states PC ('participation coefficient?'), but the legend states it is the cross-correlation?

- Many of the analyses, e.g. Fig 1, show the Participation Coefficient over time, but it is not clear to me how the time-varying PCs are calculated. Is there a sliding time window used here, and if so what is its duration? Or is this estimated at every individual TR, and if so, how can connectivity be estimated?

- Methods and Fig 1c states a threshold of $\alpha=0.1$ (assessing null distribution at the 5th to 95th percentile) rather than the standard $\alpha=0.05$ (which would yield 2.5th to 97.5th percentile). This is not consistent with typical statistical testing thresholds requiring $p<0.05$.

- Methods for 7T study should report multiband factor vs. in-plane acceleration separately; shift factor and head coil should also be reported. Methods for 3T study should state whether acceleration/multiband was used.

- Fig S2 has no axis labels, and the description is not clear, what does this represent?

Minor comments:

- fig 1 legend: 'PC' has not been defined yet, and it is confusing for the reader.

- Figure 2 legend has part f, but there's no f in the figure.

- Had to assume that $P(ds,dt)$ is calculated from data but it's not stated exactly how it was calculated.

- is fig 1d averaged across both hemispheres? or showing different hemispheres for each timepoint?

- The analysis pipeline regresses out CSF signals; however CSF signals can also be coupled to brain state. How might this affect the results? Perhaps it will not be an issue since the correlation between the ventricle and LC was also examined.

- I found the discussion section on 1st person vs. 3rd person states a bit difficult to understand; the analogy is not very clear.

Reviewer #2 (Remarks to the Author):

This paper reports on reanalysis of data from 60 younger adults' resting state activity during 7T rapid imaging. The authors took locus coeruleus (LC) and basal nucleus of Meynert (BNM) ROIs and identified time points during the resting scan when one of these nuclei showed a larger increase in BOLD signal than the other. Cross-correlations of the LC-BNM values with time series from 400 cortical regions showed that frontal regions responded faster (about 1.5 s later) than posterior visual regions (around 3-4 s later). In addition, after a greater LC than BNM increase, the ventral attention network, executive control and default mode networks showed greater correlated activity within and between networks. In contrast, somatosensory networks showed more within-network correlations before than after a LC-BNM increase (a finding not remarked upon by the authors). Additional analyses were conducted using a second previously published 3T dataset (N=14; with TR = 1500 ms) consisting of experienced meditators scanned while focusing on their breath and pressing a button whenever they realized their mind had wandered. Here LC>BNM activity was modeled in the 5 TRs prior to and following each button press. Participants showed a peak in τ LC-BNM and in mean correlated activity (participation coefficient) around 4 seconds before they pressed the button.

This is an innovative approach with a rich set of results that are likely to have a significant impact in the field. In my opinion, it is the most sophisticated and convincing approach to extracting meaningful LC signal from fMRI to date. I'm quite enthusiastic about the potential of this approach to help elucidate the role of neuromodulators on patterns of brain activity.

The main ways I believe this paper could be improved are: first, to take into account the full scope of the findings to better align the interpretation of the LC influences with previous findings in the literature (see #1-2 below); and second, that there are a number of outcomes/details of these analyses that were not provided that should be.

Major points

1. The authors argue that noradrenaline integrates brain activity whereas cholinergic activity segregates it ("Together, these findings provide robust evidence for the hypothesis that the balance between ascending noradrenergic and cholinergic tone facilitates a transition towards topological integration across the distributed network of the brain." p. 7). Yet their data suggest that the noradrenaline integration effects are not universal across the brain. In Figure S1, while broad-scale networks such as VAN, CON and DMN show the pattern of greater within and between network connectivity, there are other patterns that are also quite interesting. First, the two sensory networks the authors included, the

visual and the somatomotor networks, do not show this pattern. In particular, the somatomotor network shows increased internal connectivity before the LC-BNM spike. Intriguingly, limbic-sensory connectivity is highest before the LC - BNM spike, which suggests the possibility that sensory information influencing amygdala activity triggers LC. Given the resting-state nature of these scans, it makes sense that the sensory signals that would be most influential would be internal signals represented in the somatomotor network, but if these analyses were conducted for a scan of watching a movie, a testable hypothesis would be that this pattern would emerge more strongly for visual - limbic connectivity.

2. In addition to these interesting pre LC spike patterns that may relate to what is influencing the LC - BNM activity, there also is a post-LC-BNM pattern among the Vis, SM, and DAN networks (upper left quadrant of Fig. S1) that indicates a decrease in network integration and thus an increase in network segregation post LC - BNM activity. In general, this figure suggests that large-scale coordinating networks such as VAN, DMN and CON show more coordinated activity post LC-BNM signals, but that sensory regions show more segregated activity post LC-BNM signals. This pattern fits with prior findings that, under arousal, salient or high priority stimuli are attended to and remembered better whereas less salient or low priority stimuli are even more neglected than they would be otherwise in a 'winner-take-more' pattern (Mather & Sutherland, 2011), a phenomenon that has been hypothesized to be due to glutamate-beta-adrenergic 'hot spots' of amplified activity at the site of highly active sensory representations under arousal (Mather, Clewett, Sakaki & Harley, 2016). This greater gain on the competition between sensory inputs is not what would be predicted if sensory regions were showing more integrated activity. However, if sensory regions became more segregated while large-scale networks that coordinate broad-scale activity became more integrated, presumably whatever representation was most active in sensory regions would gain even more resources. While the authors may or may not agree with my interpretation of the connectivity patterns, it seems important to try to address these patterns from the sensory networks that do not fit well with their current argument.

3. A theoretically important question that should be addressed is whether the LC-BNM models account for significantly more variance than one that only models LC phasic increases in activity.

4. Relatedly, what were the results from the BNM-LC and LC+BNM models? For instance, it would be helpful to see panels Figure 1B-E and Figure S1 (that show LC-BNM model results) recreated for these other two models.

5. p. 18: More detail on how τ_{LC-BNM} , τ_{BNM-LC} and τ_{LC+BNM} are each calculated would be helpful. For instance, for τ_{LC+BNM} , were the time series of LC and BNM averaged and then a second derivative taken from that, or were separately calculated second derivatives averaged? Or were the LC and BNM phasic increases integrated later in the calculating pipeline?

6. How were the eight resting-state networks plotted in Figure S1 computed/selected? And why are some networks apparently represented by 2 rows and others by 3 or 1?

7. It is exciting to see that this approach may be feasible using 3T (3mm x 3mm x 4mm³) as well as 7T data. It would strengthen this report if there were some validation of these analyses. Much less basic information/analyses are presented for this 3T dataset than for the 7T one. One start would be to report the comparison of the LC and BNM time series to potential nuisance signals (see top of p. 18) for this data set as well. Also, how many τ LC-BNM, τ BNM-LC and τ LC+BNM time points were identified across subjects?

Other questions

It is stated that the data were originally described in Hearne et al. (2017). There is no referenced entry in the bibliography for this paper, however, it seems they are referring to Hearne, Cocchi, Zalesky, & Mattingley (2017). In this paper, 65 participants were scanned, but four were excluded due to “MR scanning issues” one to a brain structure abnormality, one to poor behavioral task performance, and 10 due to excessive head movement. In the present paper, it is simply stated that 60 of the 65 were included in the final analysis, but it is not specified the basis for exclusions. Later, the authors state that 5 participants were excluded for having head displacement > 3mm in > 5% of volumes. Then on p. 18, it seems that only 59 subjects were used. As Hearnes et al. excluded 10 for excessive head movement, the exclusion criteria used here and how it lines up with the Hearnes et al. paper exclusions should be explained. Also, in the Hearne et al. paper, it states that each participant completed two 10-min resting scans. Here it seems just one of these were used but it is not clear which of the two nor why just one was used.

Why was .071 Hz selected as the low-frequency bound of the band pass filter? Were other frequency bands modeled and if so, is this a more effective one? Was this preprocessing step in place when obtaining the resting-state network maps used in Figure S1?

Figure 1 - the caption specifies an “F” panel that is not included in the figure.

p. 6: “An increase in phasic activity within the LC (relative to the BNM) consistently preceded an increase in the level of integration within the cerebral cortex that was maximal in frontoparietal cortices (and v.v.; Fig. S1).” - I assume that ‘v.v.’ is short for vice versa, but I’m not sure what that means in terms of Fig. S1 (should it mean that BNM-LC shows the opposite pattern of activity?).

Does the 'CON' 'control' network refer to a PFC executive network? Please clarify as 'CON' has also been used to refer to the cingulo-opercular network.

It seems that the unit of analysis was each tLC-BNM time point. What was the mean number of these time points per subject and the range, in both of the datasets? Was subject included as a factor in the analyses to appropriately partition within- vs. between-subject variance?

It is confusing that in this manuscript and in the figures, "PC" can refer either to "Principal Component" or to "Participation Coefficient". Please either avoid the "PC" abbreviation altogether or figure out some unique abbreviations for each of these.

I have been convinced by discussions about the replicability crisis that it should be a standard practice to share data upon publication. Please address whether you have prepared your data, scripts and materials for release upon publication in a publicly accessible online repository such as OpenNEURO or Open Science Framework. If so, please provide the URL, DOI, or other permanent path for accessing the data in a public, open access repository. Making data available can increase the impact of the research study. Simply stating "data available upon request" is not sufficient as, unfortunately, when data are not shared upon publication in a public repository, it can be impossible to gain access by request from the researchers — the majority of requests from other researchers to obtain data are not complied with (e.g., Vanpaemel, Vermorgen, Deriemaecker, & Storms, 2015; Wicherts, Bakker, & Molenaar, 2011).

Mara Mather

(I sign all reviews)

Reviewer #3 (Remarks to the Author):

In this work, the authors investigated how neurochemical substances released from the brain stem affect brain dynamics and network topology. I think the theme is important and the datasets obtained by a 7T MRI give some methodological significance to this work. In the meantime, the analysis methods, including the definition of the energy value and the concept of energy landscape, are not sufficiently logical or mathematically supported, and some final claims based on the results do not sound well convincing. Also, I am afraid that the manuscript does not contain enough information about the statistics (e.g., the names of the tests, degree of freedom, effect size and, if necessary, correction for multiple comparisons). Given this, it is difficult for me to recommend the current form of the manuscript for the publication in Nature Communications.

- Not-yet-validated association between energy value and probability

In almost all cases of physics studies using energy landscape concept, the occurrence probability of a certain state, P , is defined as $P = 1/Z * \exp(-E/T)$, where E is the energy value for the specific state, T denotes a temperature and Z is a normalisation constant. In a line of previous neuroscience work using this concept, the mathematical association between the energy value and the probability was the same as this or at least qualitatively the same (like just set $T = 1$ in some studies).

However, this study appears to set their own equation, based on which the appearance probability was converted into the energy value.

Given a large part of the current findings relies on this equation, I think the authors may have to provide some solid logic to justify this energy-probability relationship.

- Unclear description of the way to calculate the probability

At least for me, how to calculate the occurrence probability—a critical value for this research—is not so much clear. The probability of the MSD of the, in my understanding, certain whole-brain activity pattern seems to be calculated based on a specific time window with a length of dt TR; but it is obscure how to calculate such an appearance frequency in a non-binarised parametric space. It'd be quite helpful if the authors elaborate this point.

- Energy landscape or energy landscape of energy landscape?

If the definition of the energy value is justified and the way to estimate the occurrence probability is validated, this study still seems to have a major energy-landscape-related issue to be solved.

In my understanding, the energy value used here seems to be inversely correlated with the stability of a certain whole-brain activity pattern and thus the resultant energy landscape should be a foundation of brain state dynamics during the observed time window. Therefore, in this sense, it is reasonable to assume that “attractors” and “brain states” are those on such an energy landscape and represent groups of brain activity patterns.

However, the authors seem to use these terms differently. As described in the sentences bridging between page nine and ten, the authors use an attractor as dips in the time-varying MSD changes (Figure 2C), which looks sort of dynamics of energy landscape structures and sounds discussion of energy landscape of energy landscape.

In my feeling, it would be helpful if the authors clarify this issue or use different terms for clearer and less confusing discussions.

- Difficult to specify conventional energy landscape structures

In addition, the current analysis framework does not seem to allow the authors to specify (conventional) attractors, which gives some vagueness to the interpretation of the main results (Figure 2C).

This study used MSD as an index for brain activity patterns. Theoretically, the MSD contains different and multiple brain activity patterns; therefore, we cannot specify which brain activity patterns make attractors with which other activity patterns just based on Figure 2C and Figure S3. (That's also why I felt some concerns about discussions/interpretations about "attractors" in this manuscript...)

Also, by the same logic, it is difficult to assure that the "Reaction path" in the Figure 3B is the same across the three different conditions.

Therefore, in my understanding, the current analysis seems to have difficulty in estimating details of structural changes in the energy landscapes (e.g., the size of attractors and the height of energy barriers). This means that the current findings appear to be not the direct but circumstantial evidence for the main claim of this study (i.e., neurochemicals from the brain stem affect energy landscapes)...

I hope the authors will find out some creative ways to address this concern.

- Validation for τ in the mediator experiment.

In the additional experiment employing 14 expert mediators, the authors used a 3T MRI. In my experience, it would be somewhat difficult to accurately infer brain stem activity in a 3T MRI scanning. So it'd be nice if the authors give us some evidence that, as in the main experiment, validates the brain stem activity recorded in the 3T fMRI (in particular, τ value presented in Figure 3).

- Inconsistent description of sample size

It may be just a minor issue but I think the authors stated different numbers for the participants whose data were put into final analysis. In the first paragraph of the Methods section, they stated "60 were included in the final analysis", whereas the "Phasic increases in neuromodulatory BOLD signal" section read "...across all 59 subjects".

- Justification for the sample sizes

This study induced a relatively large number of participants for the main experiment ($N \sim 60$) and a relatively small sample size for the additional analysis of the fMRI data recorded from the mediators ($N \sim 15$). So, I think it may be useful if the authors stated any justifications for such seemingly diverse sample sizes (e.g., power analysis, preferably).

- Where is the panel F in Figure 1?

The figure legend has a sentence for an invisible panel F...

REVIEWER COMMENTS

Reviewer #1 (Remarks to the Author):

Major comments:

- The study centers on interpretation of LC signals, but uses 2mm voxels. Since LC is only 1-2 mm in width, this suggests that other structures were present in the ROI. While supresolution voxel sizes can be used successfully in certain structures with shapes that allow for averaging out the effects of the neighbouring structures, this is not the case for the LC. The paper should either show that similar dynamics are not obtained when studying signals from immediately adjacent voxels in those nearby structures, or should acknowledge/discuss that the measured signals reflect activity from LC and its neighbouring structures, and not overstate the specificity of the imaging.

Response:

We thank the reviewer for this comment. In the original manuscript, we attempted to correct for this issue in the locus coeruleus by regressing the dynamics of an ROI located in the nearby fourth ventricle. However, as the Reviewer mentions, this would not account for any fluctuations shared across nearby voxels comprised predominantly of signals from grey matter structures. To avail this concern, we extracted the mean activity of the locus coeruleus mask after shifting the mask anteriorly such that it overlapped with an area that harbours the pontine nuclei (i.e., +8mm in the Y direction). In the same manner in which we previously regressed the dynamics of the fourth ventricle, we regressed the activity of this non-LC pontine region, and then re-analysed our data. As demonstrated below, after applying this regression, the new τ_{LC} is identical to the original LC time series (i.e., $r = 1$). As such, each of the results was statistically identical following this approach, providing confidence that the original conclusions were not biased by a lack of regional specificity.

The following text was added to the manuscript (p18):

“To ensure that BOLD signals from nearby grey matter structures were not influencing the locus coeruleus timeseries, we extracted the mean activity of the locus coeruleus mask after shifting the mask anteriorly such that it overlapped with an area that harbours the pontine nuclei (i.e., +8mm in the Y direction). In the same manner in which we previously regressed the dynamics of the fourth ventricle, we regressed the activity of this non-LC pontine region, and then re-analysed our data. Each of the results was statistically identical following this approach, providing confidence that the original conclusions were not biased by a lack of regional specificity.”

- It would also be helpful to show an overlay of selected functional voxels on the anatomical localization of LC, describe the probability threshold used in the ROI definition, and report the number of voxels in the ROI.

Response:

As requested, we have added a new figure (S1) to the Supplementary Materials.

Fig S1. Locus coeruleus template. Left: The anatomical locus coeruleus mask projected onto MNI 0.5mm standard brain (orange); Right: The anatomical locus coeruleus mask down-sampled onto an example 7T Echo Planar Image from a single subject (red).

- Due in part to this issue, some of the conclusions seem overstated. E.g., “The results of our initial analysis confirm that distributed activity in the brain is modulated by small groups of neuromodulatory cells in the brainstem and forebrain” after Fig. 1. I agree that the results are interesting, but this result does not show that neuromodulatory balance causally controls brain activity, particularly since LC is tiny and the LC voxel surely contained other structures as well – rather, this is already known to be true from animal studies, and this paper provides an interesting new analysis of that distributed activity. Slightly more cautious statements could be used throughout the results and discussion.

Response:

We agree with the Reviewer and have diminished our claims in line with the evidence obtained. We have modified the following sentences:

The following text was added to the manuscript (p7):

“The results of our initial analysis demonstrate that coordinated distributed activity in the cortex align with changes in small groups of neuromodulatory cells in the brainstem and forebrain, which in turn are proposed to constrain brain dynamics onto a low-dimensional energy landscape (Fig. 2A).”

The following text was added to the manuscript (p13):

“Our results ~~confirm~~ support this prediction by showing that patterns of activity in key regions within the brainstem”

The following text was added to the manuscript (p16):

“Our results ~~confirm~~ support specific predictions from a recent hypothetical framework¹⁵”

- There are no statistics for the lags across cortical regions, and it would be good to show whether at least earliest and latest are significantly different.

Response:

We apologise for the lack of clarity. We did indeed run a parametric block-resampling approach to create a null model that preserved autocorrelation while scrambling the precise sequence inherent within the data. The information is presented on p7 and p21 of the manuscript.

- Further explanation of the baseline dynamics would be helpful. Why are the energy landscapes in Figure 2C control shaped the way it is? Why does it decrease around 10? This would make the energy landscapes of LC-BNM and BNM-LC more meaningful.

Response:

We have included the following sentences into the results when introducing the MSD energy landscapes.

The following text was added to the manuscript (p10):

“These figures demonstrate an MSD energy landscape across displacement and time, wherein the energy relates to the likelihood of seeing a given mean change in bold activity (i.e., MSD) at a given temporal displacement (i.e., TR). For example, all the MSD energy landscapes have a high energy peak for large MSD at a short timescale as it is extremely unlikely that the bold activity would change significantly (quantified by a large MSD) in one TR (~0.5s), and neuromodulation increases the energy of such an initial change. The utility of the MSD energy landscape can be seen when comparing large phasic bursts of LC and BNM relative to baseline fluctuations. We found the largest change occurs around 10-15 TR (~6-9s) following a phasic burst that typically corresponds to a peak in the LC or BNM BOLD signal. At this ~6-9s temporal delay we see direct evidence that a phasic burst of LC flattened the energy landscape (relative to baseline Fig. 2C, red inset), thus making previously unlikely large MSD trajectories far more probable, whereas a phasic burst of BNM activity (relative to baseline; Fig. 2C, green inset) caused the energy landscape to be elevated, thus promoting local trajectories, and making large MSD deviations unlikely.”

- Fig 1C y-axis label states PC (‘participation coefficient’?), but the legend states it is the cross-correlation?

Response:

We apologise for this confusion. The y-axis is the cross-correlation between τ_{LC-BNM} and the PC time series and we have updated the figure to specify this.

The following text was added to the manuscript (p6):

“As predicted, we observed a strong positive ~~relationship~~ correlation between τ_{LC-BNM} and mean network-level integration ($p < 0.05$; Fig. 1C).”

- Many of the analyses, e.g. Fig 1, show the Participation Coefficient over time, but it is not clear to me how the time-varying PCs are calculated. Is there a sliding time window used here, and if so what is its duration? Or is this estimated at every individual TR, and if so, how can connectivity be estimated?

Response:

We apologise, as there was a missing section in our Methods. We have added this information to the updated manuscript.

The following text was added to the manuscript (p19):

“Time-resolved functional connectivity

To estimate functional connectivity between the 400 regions of interest, we used the multiplication of temporal derivatives (MTD) technique. Briefly, MTD is computed by calculating the point-wise product of temporal derivative of pair-wise time series. The resultant score is then averaged over a temporal window, w (a window length of 20 TRs was used in this study, though results were consistent for $w = 10-50$ TRs)."

- Methods and Fig 1c states a threshold of $\alpha=0.1$ (assessing null distribution at the 5th to 95th percentile) rather than the standard $\alpha=0.05$ (which would yield 2.5th to 97.5th percentile). This is not consistent with typical statistical testing thresholds requiring $p<0.05$.

Response:

We thank the reviewer for pointing out this mistake. The manuscript should have read “ $\alpha = 0.05$ ”, which corresponds to a 95-percentile range between the 2.5th and 97.5th percentiles of the null distribution. We have updated the figures and text to reflect this.

- Methods for 7T study should report multiband factor vs. in-plane acceleration separately; shift factor and head coil should also be reported. Methods for 3T study should state whether acceleration/multiband was used.

Response:

We apologise for this oversight. An acceleration factor of 5 was used in collection of the 7T functional data. Further information can be found in the original study (Hearne et al., 2017 *J Neurosci.*).

- Fig S2 has no axis labels, and the description is not clear, what does this represent?

Response:

Figure S2 has been updated.

Minor comments:

- Fig 1 legend: ‘PC’ has not been defined yet, and it is confusing for the reader.

Response:

This has been updated in the manuscript.

- Figure 2 legend has part f, but there’s no f in the figure.

Response:

We apologise for this oversight; it has been removed.

- Had to assume that $P(ds,dt)$ is calculated from data but it's not stated exactly how it was calculated.

Response:

The methods now describe this calculation in greater detail.

The following text was added to the manuscript (p21):

“We are interested in the probability, P_{MSD} , that we will observe a given displacement in BOLD at a given time-lag t . We estimated the probability distribution function $P(MSD, t)$ from n MSD_{t,t_0} samplings, – e.g., the identified n phasic bursts of subcortical structures (as above) – using a Gaussian kernel density estimation $P(MSD, t) = \frac{1}{4n} \sum_{i=1}^n K\left(\frac{MSD_{t,i}}{4}\right)$, where $K(u) = \frac{1}{2\sqrt{\pi}} e^{-\frac{1}{2}u^2}$ and we display the results for t between 1 to 15 TR and MSD between 0 to 50.”

- is fig 1d averaged across both hemispheres? or showing different hemispheres for each timepoint?

Response:

We apologise for the lack of clarity. Fig 1D only shows the right hemisphere – the “Pre” image is the right hemisphere in its original format, whereas the “Post” image is a mirrored reflection for comparison. It is worth noting that the pattern was highly similar across both hemispheres.

The following text was added to the manuscript (p6):

“(D) mean cortical participation coefficient (PC) preceding (left) and following (right) the zero-lagged τ_{LC-BNM} value, right hemispheres shown (the “Post” image is reflected);”

- The analysis pipeline regresses out CSF signals; however CSF signals can also be coupled to brain state. How might this affect the results? Perhaps it will not be an issue since the correlation between the ventricle and LC was also examined.

Response: We have regressed CSF via the ventricles and leave the interrogation of these relationships to future studies.

- I found the discussion section on 1st person vs. 3rd person states a bit difficult to understand; the analogy is not very clear.

Response:

We have updated the discussion, which now reads.

The following text was added to the manuscript (p13):

“Much in the same way that there are many different reference frames for navigation – e.g., egocentric (i.e., straight, left, right directions), which is independent of the environment; and allocentric (i.e., following compass directions and visual cues), which is dependent on the environment – we can interrogate energy landscapes using different vantage points on BOLD dynamics. Our displacement framework is consistent with an egocentric (or ‘first-person’) frame of reference, wherein MSD is used to track BOLD trajectories from an initial state which maps out the topology of the energy landscape (i.e., a BOLD MSD implies a BOLD trajectory). Nonetheless, the method does not distinguish between two different neural trajectories that possess the same MSD. In comparison, other methods have evaluated the energy landscape for a given pre-defined state estimated from thresholded BOLD timeseries^{22,23} a framework consistent with an allocentric (or ‘third-person’) reference frame. This framework has the advantage of calculating energy for a given state, however, it also requires substantial exploration of the state-space – which is typically unfeasible – or the need to resort to severe coarse-graining (such as the binarization of BOLD activity) which further diminishes interpretability. Furthermore, the allocentric view does not provide insights into the transitions between each energy state, whereas this information is inherent to the egocentric reference. Along these lines, we found that the egocentric reference frame clearly demonstrated the flattening and deepening of the energy landscape, providing indirect evidence that the ascending arousal system is well set-up to control brain-state dynamics ‘egocentrically’ (as opposed to specific neural activity patterns). Nevertheless, given improvements in recording length and novel analytic techniques to probe the brains dynamical landscape, we expect that the field will ultimately discover even more optimal mappings between neurobiology and low-dimensional brain state dynamics.”

Reviewer #2 (Remarks to the Author):

Major points

1. The authors argue that noradrenaline integrates brain activity whereas cholinergic activity segregates it (“Together, these findings provide robust evidence for the hypothesis that the balance between ascending noradrenergic and cholinergic tone facilitates a transition towards topological integration across the distributed network of the brain.” p. 7). Yet their data suggest that the noradrenaline integration effects are not universal across the brain. In Figure S1, while broad-scale networks such as VAN, CON and DMN show the pattern of greater within and between network connectivity, there are other patterns that are also quite interesting. First, the two sensory networks the authors included, the visual and the somatomotor networks, do not show this pattern. In particular, the somatomotor network shows increased internal connectivity before the LC-BNM spike. Intriguingly, limbic-sensory connectivity is highest before the LC - BNM spike, which suggests the possibility that sensory information influencing amygdala

activity triggers LC. Given the resting-state nature of these scans, it makes sense that the sensory signals that would be most influential would be internal signals represented in the somatomotor network, but if these analyses were conducted for a scan of watching a movie, a testable hypothesis would be that this pattern would emerge more strongly for visual - limbic connectivity.

Response:

We thank the Reviewer for this comment, and we agree that repeating this analysis during movie watching is a particularly interesting idea (and is in fact one that we're actively working on).

The following text was added to the manuscript (p6-7):

“An increase in phasic activity within the LC (relative to the BNM) preceded an increase in the mean level of integration within the cerebral cortex that was dominated by the frontoparietal cortices (Fig. 1D; parcellated according to the 17 resting-state networks identified in³¹). Interestingly, this global integration was opposed by a relative topological segregation of limbic, visual, and motor cortices (Fig. S2).”

“Together, these findings provide robust evidence for the hypothesis that the balance between ascending noradrenergic and cholinergic tone facilitates a transition towards topological integration across the ~~distributed~~ frontoparietal network of the brain”

2. In addition to these interesting pre LC spike patterns that may relate to what is influencing the LC - BNM activity, there also is a post-LC-BNM pattern among the Vis, SM, and DAN networks (upper left quadrant of Fig. S1) that indicates a decrease in network integration and thus an increase in network segregation post LC - BNM activity. In general, this figure suggests that large-scale coordinating networks such as VAN, DMN and CON show more coordinated activity post LC-BNM signals, but that sensory regions show more segregated activity post LC-BNM signals. This pattern fits with prior findings that, under arousal, salient or high priority stimuli are attended to and remembered better whereas less salient or low priority stimuli are even more neglected than they would be otherwise in a ‘winner-take-more’ pattern (Mather & Sutherland, 2011), a phenomenon that has been hypothesized to be due to glutamate-beta-adrenergic ‘hot spots’ of amplified activity at the site of highly active sensory representations under arousal (Mather, Clewett, Sakaki & Harley, 2016). This greater gain on the competition between sensory inputs is not what would be predicted if sensory regions were showing more integrated activity. However, if sensory regions became more segregated while large-scale networks that coordinate broad-scale activity became more integrated, presumably whatever representation was most active in sensory regions would gain even more resources. While the authors may or may not agree with my interpretation of the connectivity patterns, it seems important to try to address these patterns from the sensory networks that do not fit well with their current argument.

Response:

We thank the Reviewer for the insight, and we have included the referenced viewpoint into the paper

The following text was added to the manuscript (p7):

“This increase in the synchronisation of the frontoparietal cortices following an increase in sensory-limbic coordination and LC activity may reflect arousal-enhanced processing of sensory stimuli^{32,33}.”

3. A theoretically important question that should be addressed is whether the LC-BNM models account for significantly more variance than one that only models LC phasic increases in activity.

Response:

τ_{LC} and τ_{LC-BNM} both account for effectively equivalent explained variance, 8% and 6% respectively, that are both significant ($p < 0.05$; 95% CI).

4. Relatedly, what were the results from the BNM-LC and LC+BNM models? For instance, it would be helpful to see panels Figure 1B-E and Figure S1 (that show LC-BNM model results) recreated for these other two models.

Response:

Figures 1: B-E are calculated from cross-correlation between τ_{LC-BNM} and as $\tau_{BNM-LC} = -\tau_{LC-BNM}$ the opposite effects control BNM-LC and from Fig.S2 we expect LC+BNM to be dominated by the BNM-LC effect.

The following text was added to the manuscript (p7):

“These results can be inverted for BNM activity (relative to LC) as $\tau_{BNM-LC} = -\tau_{LC-BNM}$.”

5. p. 18: More detail on how τ_{LC-BNM} , τ_{BNM-LC} and τ_{LC+BNM} are each calculated would be helpful. For instance, for τ_{LC+BNM} , were the time series of LC and BNM averaged and then a second derivative taken from that, or were separately calculated second derivatives averaged? Or were the LC and BNM phasic increases integrated later in the calculating pipeline?

Response:

We apologise for the lack of clarity. This information has been refined in the updated manuscript.

The following text was added to the manuscript (p9):

“Briefly, we identified phasic bursts as peaks in the second-derivative of the arousal BOLD signals that lead to a sustained increase in arousal BOLD for each individual (see Methods).”

6. How were the eight resting-state networks plotted in Figure S1 computed/selected? And why are some networks apparently represented by 2 rows and others by 3 or 1?

Response:

These represent 17 resting-state networks from Yeo et al., 2011, *J Neurophysiol.*

We have modified the manuscript to state this clearly:

The following text was added to the manuscript (p6):

“An increase in phasic activity within the LC (relative to the BNM) preceded an increase in the mean level of integration within the cerebral cortex that was dominated by the frontoparietal cortices (Fig. 1D; parcellated according to the 17 resting-state networks identified in³¹).”

7. It is exciting to see that this approach may be feasible using 3T (3mm x 3mm x 4mm3) as well as 7T data. It would strengthen this report if there were some validation of these analyses. Much less basic information/analyses are presented for this 3T dataset than for the 7T one. One start would be to report the comparison of the LC and BNM time series to potential nuisance signals (see top of p. 18) for this data set as well.

Response:

We thank the Reviewer for this comment. In the original manuscript, we attempted to correct LC BOLD dynamics from this issue by regressing the dynamics of an ROI located in the nearby fourth ventricle. However, as the reviewer mentions, this would not account for any fluctuations shared across nearby voxels comprised predominantly of signals from grey matter structures. To avail this concern, we extracted the mean activity of the locus coeruleus mask after shifting the mask anteriorly such that it overlapped with an area of the pons that harbours the nuclei (i.e., +8mm in the *Y* direction). In the same manner in which we previously regressed the dynamics of the fourth ventricle, we regressed the activity of this non-LC pontine region, and then re-analysed our data. As demonstrated below, after applying this regression the new τ_{LC} is identical to the original LC time series. As such, each of the results was statistically identical following this approach, providing confidence that the original conclusions were not biased by a lack of regional specificity.

The following text was added to the manuscript (p18):

“To ensure that BOLD signals from nearby grey matter structures were not influencing the locus coeruleus timeseries, we extracted the mean activity of the locus coeruleus mask after shifting the mask anteriorly such that it overlapped with an area of the pons that harbours the nuclei (i.e., +8mm in the Y direction). In the same manner in which we previously regressed the dynamics of the fourth ventricle, we regressed the activity of this non-LC pontine region, and then re-analysed our data. Each of the results was statistically identical following this approach, providing confidence that the original conclusions were not biased by a lack of regional specificity.”

Also, how many τ_{LC-BNM} , τ_{BNM-LC} and τ_{LC+BNM} time points were identified across subjects?

Response:

We state the number of points identified in the methods, we identified 148 τ_{LC-BNM} time points, 130 τ_{BNM-LC} time points and 316 τ_{LC+BNM} time points across all 59 subjects and we also now state in the results the number of phasic peaks identified.

The following text was added to the manuscript (p9):

“Using these criteria, we identified 148 τ_{LC-BNM} time points and 130 τ_{BNM-LC} time points across all 59 subjects.”

It is stated that the data were originally described in Hearne et al. (2017). There is no referenced entry in the bibliography for this paper, however, it seems they are referring to Hearne, Cocchi, Zalesky, & Mattingley (2017). In this paper, 65 participants were scanned, but four were excluded due to “MR scanning issues” one to a brain structure abnormality, one to poor behavioral task performance, and 10 due to excessive head movement. In the present paper, it is simply stated that 60 of the 65 were included in the final analysis, but it is not specified the basis for exclusions. Later, the authors state that 5 participants were excluded for having head

displacement > 3mm in > 5% of volumes. Then on p. 18, it seems that only 59 subjects were used. As Hearnese et al. excluded 10 for excessive head movement, the exclusion criteria used here and how it lines up with the Hearnese et al. paper exclusions should be explained. Also, in the Hearnese et al. paper, it states that each participant completed two 10-min resting scans. Here it seems just one of these were used but it is not clear which of the two nor why just one was used.

Response:

In our analysis, we analysed 59 individuals – 4 were removed for excessive head movements, one was removed due to abnormal brain structure and one was removed for inconsistent voxel activity after global-signal regression. We only analysed one session from each individual. We have clarified the inconsistent explanation in the Methods.

The following text was added to the manuscript (p17):

“Sixty-five healthy, right-handed adult participants (mean, 23.35 years; SD, 3.6 years; range 18–33 years; 28 females) were recruited, of whom 59 were included in the final analysis (four participants were excluded due to MR scanning issues, one participant was excluded due to an unforeseen brain structure abnormality, and one was excluded due to inconsistent BOLD dynamics following global-signal regression).”

Why was .071 Hz selected as the low-frequency bound of the band pass filter? Were other frequency bands modeled and if so, is this a more effective one? Was this preprocessing step in place when obtaining the resting-state network maps used in Figure S1?

Response:

This was a typographic error. The frequency band analysed was 0.01-0.15 Hz. This information has been updated in the manuscript.

Figure 1 - the caption specifies an “F” panel that is not included in the figure.

Response:

We thank the reviewer for pointing this out and we have removed the caption.

p. 6: “An increase in phasic activity within the LC (relative to the BNM) consistently preceded an increase in the level of integration within the cerebral cortex that was maximal in frontoparietal cortices (and v.v.; Fig. S1).” - I assume that ‘v.v.’ is short for vice versa, but I’m not sure what that means in terms of Fig. S1 (should it mean that BNM-LC shows the opposite pattern of activity?).

Response:

We agree this sentence was unclear. Your interpretation is correct that BNM-LC shows the opposite pattern.

As above we have stated (p7):

“These results can be inverted for BNM activity (relative to LC) as $\tau_{BNM-LC} = -\tau_{LC-BNM}$.”

Does the ‘CON’ ‘control’ network refer to a PFC executive network? Please clarify as ‘CON’ has also been used to refer to the cingulo-opercular network.

Response:

Con refers to frontoparietal control network (Yeo et al., 2011).

We have updated the text and Figure 1 caption now reads (p5):

“CON – frontoparietal control”

It seems that the unit of analysis was each τ LC-BNM time point. What was the mean number of these time points per subject and the range, in both of the datasets? Was subject included as a factor in the analyses to appropriately partition within- vs. between-subject variance?

Response:

No, it was not as we were interested in a cross-subject effects. We agree that interindividual effects are particularly interesting, and we hope to investigate this with new datasets in future studies.

It is confusing that in this manuscript and in the figures, “PC” can refer either to “Principal Component” or to “Participation Coefficient”. Please either avoid the “PC” abbreviation altogether or figure out some unique abbreviations for each of these.

Response:

We thank the Reviewer for pointing this out and we have removed mention of principal component from the paper.

I have been convinced by discussions about the replicability crisis that it should be a standard practice to share data upon publication. Please address whether you have prepared your data, scripts and materials for release upon publication in a publicly accessible online repository such as OpenNEURO or Open Science Framework. If so, please provide the URL, DOI, or other permanent path for accessing the data in a public, open access repository. Making data available can increase the impact of the research study. Simply stating “data available upon request” is not sufficient as, unfortunately, when data are not shared upon publication in a public repository, it can be impossible to gain access by request from the researchers — the

majority of requests from other researchers to obtain data are not complied with (e.g., Vanpaemel, Vermorgen, Deriemaecker, & Storms, 2015; Wicherts, Bakker, & Molenaar, 2011).

Response:

We too strongly believe in Open Science. Indeed, all of the code required to conduct the analysis is already on github (github.com/Bmunnn/BSI), including the LC and BNM activity and the BOLD data is available from Hearne et al 2017.

The following text was added to the manuscript (p22):

“Data availability

The BOLD data was obtained from (Hearne et al., 2017)⁶³ and the subcortical timeseries (τ_{LC} and τ_{BNM}) are available at (github.com/Bmunnn/BSI).

Code availability

All the code required to conduct the analysis can be found on Github at (github.com/Bmunnn/BSI).”

Reviewer #3 (Remarks to the Author):

- Not-yet-validated association between energy value and probability. In almost all cases of physics studies using energy landscape concept, the occurrence probability of a certain state, P, is defined as $P = 1/Z * \exp(-E/T)$, where E is the energy value for the specific state, T denotes a temperature and Z is a normalisation constant. In a line of previous neuroscience work using this concept, the mathematical association between the energy value and the probability was the same as this or at least qualitatively the same (like just set $T = 1$ in some studies). However, this study appears to set their own equation, based on which the appearance probability was converted into the energy value. Given a large part of the current findings relies on this equation, I think the authors may have to provide some solid logic to justify this energy-probability relationship.

Response:

The reviewer is correct in that many neuroscience applications of energy landscapes analysis utilise the formula $P_{\sigma} = \frac{1}{Z} e^{\left(-\frac{E_{\sigma}}{T}\right)}$ from statistical mechanics (most commonly seen in thermodynamics) and the clear logic that follows a century of studies in this field. However, the reviewer is mistaken as we do use this formula, which can be shown through some trivial re-arrangement.

Briefly,

$$P_{\sigma} = \frac{1}{Z} e^{\left(\frac{-E_{\sigma}}{T}\right)}$$

where P_{σ} is the probability of a given state, σ , Z is the partition function a normalisation that ensures the probability of all possible states = 1, T is the temperature, and E_{σ} is the energy of the given state.

For our case we are interested in E_{σ} given P_{σ} , furthermore by construction $\sum_{\sigma} P_{\sigma} = 1$, thus, $Z = 1$ and we can trivially set $T = 1$, now rearranging

$$P_{\sigma} = e^{(-E_{\sigma})}$$

$$E_{\sigma} = \ln \frac{1}{P_{\sigma}} .$$

Thus, the energy landscape we are calculating is the same as the formula suggested by the Reviewer, and values are bound between $P_{\sigma}: (0, 1) \rightarrow 1/P_{\sigma}: (1, \infty) \rightarrow E_{\sigma}: (0, \infty)$. Furthermore, as $\ln x$ is monotonic and positive for $x > 1$, it logarithmically rescales the probabilities and following this simple logic: $E_{\sigma} \propto \frac{1}{P_{\sigma}}$. That is to say, the energy of a given state is proportional to its inverse probability. This result intuitively makes sense as we expect a state that is seen often to be a low energy state. We believe this explanation should clarify the logic of our analysis, however for more information please see a lovely paper with detailed supplementary details following the energy landscape approach (Tkacik et al., 2015; PNAS).

The following text was added to the manuscript (p21):

“As is typical in statistical mechanics the energy of a given state, E_{σ} , and its probability are related $P(\sigma) = \frac{1}{Z} e^{-\frac{E_{\sigma}}{T}}$, where Z is the normalisation function and T is a scaling factor equivalent to temperature in thermodynamics²⁰. In our analysis $\sum_{\sigma} P_{\sigma} = 1$ by construction and we can set $T = 1$ for the observed data. Thus, the energy of each BOLD MSD for a given at a given time-lag t, E , is then equal to the natural logarithm of the inverse probability, $P(\text{MSD}, t)$, of its occurrence: $E = \ln \frac{1}{P(\text{MSD}, t)}$.”

- Unclear description of the way to calculate the probability. At least for me, how to calculate the occurrence probability—a critical value for this research—is not so much clear. The probability of the MSD of the, in my understanding, certain whole-brain activity pattern seems to be calculated based on a specific time window with a length of dt TR; but it is obscure how to calculate such an appearance frequency in a non-binarised parametric space. It'd be quite helpful if the authors elaborate this point.

Response:

The MSD probability distribution was estimated from the large phasic burst samplings for τ_{LC-BNM} and τ_{BNM-LC} using a Gaussian kernel density estimation. A significant benefit of this technique is that it avoids the need for excessive coarse-graining and/or binarization.

The following text was added to the manuscript (p21):

“We are interested in the probability, P_{MSD} , that we will observe a given displacement in BOLD at a given time-lag t . We estimated the probability distribution function $P(MSD, t)$ from n MSD_{t,t_0} samplings, – e.g., the identified n phasic bursts of subcortical structures (as above) – using a Gaussian kernel density estimation $P(MSD, t) = \frac{1}{4n} \sum_{i=1}^n K\left(\frac{MSD_{t,i}}{4}\right)$, where $K(u) = \frac{1}{2\sqrt{\pi}} e^{-\frac{1}{2}u^2}$ and we display the results for t between 1 to 15 TR and MSD between 0 to 50.”

- Energy landscape or energy landscape of energy landscape? If the definition of the energy value is justified and the way to estimate the occurrence probability is validated, this study still seems to have a major energy-landscape-related issue to be solved. In my understanding, the energy value used here seems to be inversely correlated with the stability of a certain whole-brain activity pattern and thus the resultant energy landscape should be a foundation of brain state dynamics during the observed time window. Therefore, in this sense, it is reasonable to assume that “attractors” and “brain states” are those on such an energy landscape and represent groups of brain activity patterns. However, the authors seem to use these terms differently. As described in the sentences bridging between page nine and ten, the authors use an attractor as dips in the time-varying MSD changes (Figure 2C), which looks sort of dynamics of energy landscape structures and sounds discussion of energy landscape of energy landscape. In my feeling, it would be helpful if the authors clarify this issue or use different terms for clearer and less confusing discussions.

Difficult to specify conventional energy landscape structures. In addition, the current analysis framework does not seem to allow the authors to specify (conventional) attractors, which gives some vagueness to the interpretation of the main results (Figure 2C).

Response:

To avoid misinterpretations of readers applying prior notions of energy landscapes and the ambiguous term of attractors, we have removed the phrase ‘attractor landscape’ from the manuscript and describe the energy landscape to avoid any confusion.

This study used MSD as an index for brain activity patterns. Theoretically, the MSD contains different and multiple brain activity patterns; therefore, we cannot specify which brain activity patterns make attractors with which other activity patterns just based on Figure 2C and Figure S3. (That’s also why I felt some concerns about discussions/interpretations about “attractors” in this manuscript...)

Also, by the same logic, it is difficult to assure that the “Reaction path” in the Figure 3B is the same across the three different conditions.

Therefore, in my understanding, the current analysis seems to have difficulty in estimating details of structural changes in the energy landscapes (e.g., the size of attractors and the height of energy barriers). This means that the current findings appear to be not the direct but circumstantial evidence for the main claim of this study (i.e., neurochemicals from the brain stem affect energy landscapes)...

I hope the authors will find out some creative ways to address this concern.

Response:

As identified by the Reviewer, these are energy landscapes for a given BOLD MSD, and two different BOLD transitions may have the same MSD. Ultimately, as discussed throughout the paper, this approach is an alternative method to calculate energy landscapes using a different reference frame (though following identical logic, as described above). Specifically, the concern here comes from comparing to earlier energy landscape papers where a state is defined by binarising voxel activity – briefly, while this ‘allocentric’ reference frame can speak of structural changes, it is limited to states of binarised BOLD due to sampling limitations and trajectories must be inferred. Whereas, the MSD energy landscape, as an ‘egocentric’ view, does not require binarization and we can speak of subtle changes in BOLD and the energy landscape describes the ‘energy’ of a BOLD trajectory (change in BOLD activity at a given TR).

Despite the difference in reference frames all other measures of energy are the same. To address the concerns:

- **Height of barriers are then the difference in energy to transition a required BOLD MSD, and**
- **The size of an MSD state attractor is the size of the local minima in the MSD energy landscape.**

The clear raising and lowering of the energy landscape after LC and BNM phasic bursts then directly support the claim that the neuromodulatory system (from the LC and BNM) can alter the energy landscape.

- Validation for τ in the mediator experiment. In the additional experiment employing 14 expert mediators, the authors used a 3T MRI. In my experience, it would be somewhat difficult to accurately infer brain stem activity in a 3T MRI scanning. So it'd be nice if the authors give us some evidence that, as in the main experiment, validates the brain stem activity recorded in the 3T fMRI (in particular, τ value presented in Figure 3).

Response:

We agree 3T is typically very challenging for tracking subcortical structures. Nevertheless, we follow the strict protocol outlined in the methods and the fact the findings demonstrate predictions made from the first experiment above statistical significance ($p < 0.05$; 95% CI) support our assertions.

- Inconsistent description of sample size. It may be just a minor issue but I think the authors stated different numbers for the participants whose data were put into final analysis. In the first paragraph of the Methods section, they stated “60 were included in the final analysis”, whereas the “Phasic increases in neuromodulatory BOLD signal” section read “...across all 59 subjects”.

Response:

We thank the reviewer for pointing out this mistake. 59 subjects were utilised in the final analysis. From the original 65, four were removed for head movement, one was removed for structural abnormalities, and a further subject was removed for inconsistent voxel activity following GSR. This has been corrected in the methods.

- Justification for the sample sizes. This study induced a relatively large number of participants for the main experiment ($N \sim 60$) and a relatively small sample size for the additional analysis of the fMRI data recorded from the mediators ($N \sim 15$). So, I think it may be useful if the authors stated any justifications for such seemingly diverse sample sizes (e.g., power analysis, preferably).

Response:

We agree with the reviewer that the sample size of 14 is limited – unfortunately that is all the data that was available from this graciously-shared open dataset. Importantly, the task-based nature of the meditation dataset precluded the use of the same measures we calculated on the 7T resting state dataset, which thus render any potential power analysis underspecified. However, we do test for hypotheses that were inspired by findings from the first experiment and in validating these, we utilise evidence above 95% CI a $p < 0.05$ to support our claim.

- *Where is the panel F in Figure 1? The figure legend has a sentence for an invisible panel F...*

Response:

We thank the reviewer for pointing this out and we have removed the old label.

REVIEWER COMMENTS

Reviewer #1 (Remarks to the Author):

I appreciate the authors' thoughtful revisions and have just one remaining comment that they may wish to consider briefly discussing. While the addition of a new, pontine regression is helpful, it cannot fully confirm that the reported LC signals are not influenced by an immediately adjacent structure, since it is 8 mm away (whereas LC is <2 mm), and LC is surrounded by many other arousal-relevant brainstem structures, which will especially affect their 3T data. I also was surprised that the correlation was exactly 1 after regressing this region, as it seems biologically and statistically unlikely that there was exactly zero correlation between signals from the LC and the pontine nuclei. I congratulate the authors on their interesting results and hope they will consider briefly mentioning the limitation of spatial specificity in LC localization in the discussion.

Reviewer #2 (Remarks to the Author):

The authors have addressed most of the reviewer concerns but there were a couple of issues/questions they did not answer:

1. The authors addressed most but not all of my questions about how their data usage lines up with the Hearne et al. (2017) study they took it from. They state in the response that they used one session from each individual but do not state how they chose which session. To allow for replication, it would be helpful to state whether they always took the first scan or whether they had some quality criterion to pick the better of the two scans from each individual.

2. The authors did not answer my question regarding what was the mean number of τ LC-BNM time points per subject and the range (i.e., what was the minimum and maximum number per participant?). Insofar as the number of τ LC-BNM time points may have differed widely among subjects, it may be that those subjects with many τ LC-BNM time points are driving the effects.

Otherwise, this very interesting and innovative paper is in great shape.

Mara Mather

Reviewer #3 (Remarks to the Author):

Thank you for the revision. The authors have now appeared to address all my concerns except for only one but critical one.

The seemingly unaddressed or still unclear concern is about the definition of the energy. In the original manuscript, the authors defined $E = 1/P$; in the response letter, they tried to justify it, which is unfortunately not so convincing but mathematically incorrect.

First, as the authors stated (and pointed in my first review), $E = 1/\ln(P)$ is fine. And the paper the authors cited is, of course, nice as it is based on the definition.

However, $E = 1/P$ is unfortunately not precise for the $E = 1/\ln(P)$. (and the PNAS paper the authors referred to did not adopt the definition).

For example, for $P = 0.3$, $1/P = 3.3$ and $1/\ln(P) = -0.83$. For $P = 0.1$, $1/P = 10$ and $1/\ln(P) = -0.43$.

Even worse, the gap between the $1/P$ and $1/\ln(P)$ is not constant. On the contrary, as you can easily confirm, the gap shows the minimum value at $P = 0.5$ and rapidly increases at both of the ends of the $(0,1)$ period of P .

This means that we cannot correct the gap even if we add a certain constant. More importantly, the differences in the energy value between different brain states with different P values are unreliable when we adopt the $E = 1/P$ definition, which should have critical effects on energy landscape analysis, such as calculating the heights of the energy barriers.

In terms of this, the part of the response letter looks incorrect, I'm afraid.

On the other hand, the revised manuscript denoted E as $1/\ln(P)$. If the authors re-calculated all the relevant energy landscape analysis using the correct energy definition in the revised manuscript, my concern seems to have been addressed appropriately.

Nevertheless, in the response letter, unfortunately, there was no such description, and the results about the energy landscape in the revised manuscript look almost the same as those seen in the original one.

Given this, I have to say it is unclear which energy definition the authors used in the revised manuscript.

REVIEWER COMMENTS and Responses

Reviewer #1 (Remarks to the Author):

I appreciate the authors' thoughtful revisions and have just one remaining comment that they may wish to consider briefly discussing. While the addition of a new, pontine regression is helpful, it cannot fully confirm that the reported LC signals are not influenced by an immediately adjacent structure, since it is 8 mm away (whereas LC is <2 mm), and LC is surrounded by many other arousal-relevant brainstem structures, which will especially affect their 3T data. I also was surprised that the correlation was exactly 1 after regressing this region, as it seems biologically and statistically unlikely that there was exactly zero correlation between signals from the LC and the pontine nuclei. I congratulate the authors on their interesting results and hope they will consider briefly mentioning the limitation of spatial specificity in LC localization in the discussion.

Response:

We thank the reviewer for their comment. Indeed, we appreciate the importance of covarying for the potential lack of specificity in BOLD recordings of small structures in the brainstem and subcortex. As I'm sure the reviewer appreciates, the problem is very difficult to specify in such a way that all potential sources of spurious 'noise' are accounted for without potentially negatively impacting the signal in the target region. For instance, if we chose a nuisance regressor too close to the LC mask, then we would run the risk of removing the very signal we were interested in tracking. We also did not wish to collect BOLD data from predominantly descending/ascending white matter tracts, and hence avoided the middle of the pons. We hope that our reasoned approach is sufficient to demonstrate the balance between the caution with which we conducted our study and our interest in testing a specific hypothesis.

The following text was added to the manuscript (p18)

"Each of the results was statistically identical following this approach, providing confidence that the original conclusions were not biased by a lack of regional specificity. However, as the LC is surrounded by various arousal controlling nuclei, the signal likely contains BOLD activity of adjacent regions. We further expected this issue to be more significant with the reduced spatial resolution in the 3T recording, nevertheless, the similarity in the findings between the 3T and the 7T analysis provides confidence of our claims."

Reviewer #2 (Remarks to the Author):

The authors have addressed most of the reviewer concerns but there were a couple of issues/questions they did not answer:

1. The authors addressed most but not all of my questions about how their data usage lines up with the Hearne et al. (2017) study they took it from. They state in the response that they used one session from each individual but do not state how they chose which session. To allow for replication, it would be helpful to state whether they always took the first scan or whether they had some quality criterion to pick the better of the two scans from each individual.

Response:

We thank the reviewer for pointing this out. The first session was chosen for each participant. This has been made explicit in the updated manuscript.

The following text was added to the manuscript (p17)

“ These data were originally described in Hearne et al., 2017 and we selectively analysed the first resting state recording acquired from each individual⁶³ ”

2. The authors did not answer my question regarding what was the mean number of τ LC-BNM time points per subject and the range (i.e., what was the minimum and maximum number per participant?). Insofar as the number of τ LC-BNM time points may have differed widely among subjects, it may be that those subjects with many τ LC-BNM time points are driving the effects.

Response:

For the 148 significant phasic increases of LC-BNM activity > 2std, these were distributed with a mean of 2.5 per individual and a range between [0 5]. For the 130 significant phasic increases of BNM-LC activity > 2std, these were distributed with a mean of 2.2 per individual and a range between [0 7]. Finally, the 316 significant phasic increases of LC+BNM activity > 2std, these were distributed with a mean of 5.4 per individual with a range between [2 10]. Thus, at most an individual contributed ~ 5% to the effect.

The following text was added to the manuscript (p18)

“Using these criteria, we identified 148 τ_{LC-BNM} time points (mean 2.5 per individual with a range between [0 5]), 130 τ_{BNM-LC} time points (mean 2.2 per individual with a range between [0 7]) and 316 τ_{LC+BNM} time points (mean 5.4 per individual with a range between [2 10]) across all 59 subjects.”

Reviewer #3 (Remarks to the Author):

Thank you for the revision. The authors have now appeared to address all my concerns except for only one but critical one.

The seemingly unaddressed or still unclear concern is about the definition of the energy. In the original manuscript, the authors defined $E = 1/P$; in the response letter, they tried to justify it, which is unfortunately not so convincing but mathematically incorrect.

First, as the authors stated (and pointed in my first review), $E = 1/\ln(P)$ is fine. And the paper the authors cited is, of course, nice as it is based on the definition.

However, $E = 1/P$ is unfortunately not precise for the $E = 1/\ln(P)$. (and the PNAS paper the authors referred to did not adopt the definition).

For example, for $P = 0.3$, $1/P = 3.3$ and $1/\ln(P) = -0.83$. For $P = 0.1$, $1/P = 10$ and $1/\ln(P) = -0.43$.

Even worse, the gap between the $1/P$ and $1/\ln(P)$ is not constant. On the contrary, as you can

easily confirm, the gap shows the minimum value at $P = 0.5$ and rapidly increases at both of the ends of the $(0,1)$ period of P .

This means that we cannot correct the gap even if we add a certain constant. More importantly, the differences in the energy value between different brain states with different P values are unreliable when we adopt the $E = 1/P$ definition, which should have critical effects on energy landscape analysis, such as calculating the heights of the energy barriers.

In terms of this, the part of the response letter looks incorrect, I'm afraid.

On the other hand, the revised manuscript denoted E as $1/\ln(P)$. If the authors re-calculated all the relevant energy landscape analysis using the correct energy definition in the revised manuscript, my concern seems to have been addressed appropriately.

Nevertheless, in the response letter, unfortunately, there was no such description, and the results about the energy landscape in the revised manuscript look almost the same as those seen in the original one.

Given this, I have to say it is unclear which energy definition the authors used in the revised manuscript.

Response:

We apologise this wasn't clear in the first response.

The energy landscapes remain unchanged as we always used the equation $E=1/\ln(P) = \ln(1/P)$ (as can be seen in the provided Github code) and the original manuscript was missing the logarithm sign, which has been corrected.

REVIEWERS' COMMENTS

Reviewer #1 (Remarks to the Author):

The authors have addressed my comments.

Reviewer #2 (Remarks to the Author):

The reviewers have addressed my remaining concerns.

Reviewer #3 (Remarks to the Author):

The authors clarified the point, which addressed my concern.